

# Probabilistic reconstruction of sea-level changes and their causes since 1900

Sönke Dangendorf[1], Qiang Sun[1], Thomas Wahl[2], Philip Thompson[3], Jerry X. Mitrovica[4], Ben Hamlington[5]

[1]Department of River-Coastal Science and Engineering, Tulane University, 6823 St. Charles Avenue New Orleans, LA 70118, USA
[2] Department of Civil, Environmental and Construction Engineering, University of Central Florida, Orlando, FL 32816
[3]Department of Oceanography, University of Hawai`i at Manoa, Honolulu, HI, 96822
[4]Department of Earth and Planetary Sciences, Harvard University, Cambridge, MA, 02138
[5]Jet Propulsion Laboratory, California Institute of Technology, Pasadena, CA, USA

*Correspondence to*: Sönke Dangendorf (sdangendorf1@tulane.edu)

**Abstract.** Coastal communities around the world are increasingly exposed to extreme events that have been exacerbated by rising sea levels. Sustainable adaptation strategies to cope with the associated threats require comprehensive understanding of past and possible future changes. Yet, many coastlines lack accurate long-term sea level observations. Here, we introduce a novel probabilistic near-global reconstruction of relative sea-level changes and their causes over the period 1900 to 2021. The reconstruction is based on tide gauge records and incorporates prior knowledge about physical processes from ancillary observations and geophysical model outputs. We demonstrate good agreement between the reconstruction and satellite altimetry and tide gauges (if local vertical land motion is considered). Validation against steric height estimates based on independent temperature and salinity observations over their overlapping periods shows moderate to good agreement in terms of variability, though with larger trends in three out of six regions. The linear long-term trend of the resulting global mean sea level (GMSL) record is 1.5±0.19 mmyr-1 since 1900, a value consistent with central estimates from the 6th Assessment Report of the Intergovernmental Panel on Climate Change. Multidecadal trends in GMSL have varied with enhanced rates in the 1930s, near-zero rates in the 1960s, and a persistent acceleration (0.08±0.04 mmyr-2) thereafter. As a result, most recent rates have exceeded 4 mmyr-1. Largest regional rates (>10 mmyr-1) over the same period have been detected in coastal areas near western boundary currents and the larger tropical Indo-Pacific region. Barystatic mass changes due to ice-melt and terrestrial water storage variations have dominated the sea-level acceleration at global scales, but sterodynamic processes are the most crucial factor locally, particularly at low latitudes and away from major melt sources. These results demonstrate that the new reconstruction provides valuable insights into historical sea-level change and its contributing causes complementing observational records in areas where they are sparse or absent.



## 1 Introduction

Sea-level rise is one of the most impactful consequences of a warming climate threatening hundreds of millions of people in low-lying coastal areas (Hinkel et al., 2014). Many of the world's coastal areas are already actively experiencing the impacts
of rising sea levels. Along the East and Gulf coasts of the United States, rising seas have resulted in an exponential increase of chronic flooding (Sweet et al., 2014; Sweet et al., 2022; Li et al., 2023; Sun et al., 2023) affecting people's daily lives on a now regular basis. Larger than global rates of sea-level rise around the low-lying Solomon Islands in the Western Pacific led to severe shoreline recession and the submergence of several islands since the 1930s (Albert et al., 2016). For individual extreme events, such as superstorm Sandy (New York) or Hurricane Katrina (New Orleans), the underlying (anthropogenic)
sea-level rise since the late 19th century has caused significant excess flooding resulting in more severe impacts than would have occurred in an undisturbed climate (Irish et al., 2014; Strauss et al., 2021). In Africa, 20% of heritage sites of 'Outstanding Universal Value' are currently at risk of a one in a hundred-year coastal extreme event, and the number of sites is projected to triple by 2050 under continued sea-level rise (Vousdoukas et al., 2022). These examples underscore that understanding historical sea-level rise and its individual contributing factors is critically important to better project future climatic changes
and the resulting impacts along the coast.

Since 1992, radar altimeters on board of satellites have continuously monitored sea-surface height changes with near-global coverage and an accuracy of two to three centimetres locally (Ablain et al., 2015). The resulting record of global mean sea-level (GMSL) change indicates a significant increase of about 3.4 mmyr$^{-1}$ with an underlying acceleration of about 0.09 mmyr$^{-2}$ (Nerem et al., 2018). Moreover, the magnitude of the spatial variability in the rates of rise can be significantly larger in some
regions than the GMSL estimate (Wahl & Dangendorf, 2022) indicating that for local coastal planning and adaptation purposes, local sea-level information, as provided by satellites, is of utmost importance. However, the current satellite record has three major limitations. First, it is only available for 30 years, a period still too short to unambiguously detect the emergence of climatic trends and accelerations at a local level (Haigh et al., 2014). Second, sea-surface measurements are more accurate in the open ocean than close to the coast (e.g., Cazenave et al., 2020). Third, its measurements only provide information about
changes within the ocean itself (i.e., the absolute sea level) but not the coastal land. The combination of absolute sea level and vertical land motion (VLM) (i.e., the relative sea level), however, is the most relevant to decision makers in the coastal zone. Measurements of relative sea level over periods longer than the satellite record are provided by tide gauges, including some locations such as Amsterdam (van Veen, 1945), Stockholm (Ekman, 1988), or Brest that have records spanning several hundred years (Woodworth et al., 2010). Tide gauges have been measuring sea level at several thousand locations worldwide, and, the
exception of a few island locations, these records are limited to the coastal zone of the continents (Church and White, 2011). Scientists have therefore developed mathematical approaches to reconstruct the geometry of sea level from the sparse tide gauge records. One approach, which is based on Reduced Spatial Optimal Interpolation (RSOI), determines the spatial patterns of sea-surface height from satellite altimetry via empirical orthogonal functions and then fits the leading patterns to the sparse tide gauge record, providing a spatiotemporal reconstruction of sea level with the same spatial resolution as satellite altimetry





and a length similar to tide gauge records (e.g., Church et al., 2004; Church et al., 2006; Church and White, 2011; Ray and Douglas, 2011; Meyssignac et al., 2012). The second approach, which uses a Bayesian framework and ensemble Kalman smoothing (Hay et al., 2013; Hay et al., 2015; Hay et al., 2017), employs a priori known spatial patterns of individual sea-level contributors such as ice-melt (the so-called fingerprints of gravitation, rotation, and deformation, GRD), glacial isostatic adjustment (GIA), and modelled sterodynamic (i.e., the combination of steric expansion and ocean circulation changes) sea

level from climate models that are fitted to the tide gauge records. Both approaches provide GMSL estimates that agree well with satellite altimetry trends over the overlapping periods since 1993, but their GMSL estimates differ before the 1970s. The Kalman Smoother-based technique indicates a lower trend ($\sim$1.2$\pm$0.2 mmyr$^{-1}$) than the RSOI-based techniques (1.6-1.9 mmyr$^{-1}$) before 1990, thus leading to a larger acceleration over the entire 20$^{th}$ century (Hay et al., 2015). Recent assessments based on different reconstruction techniques have indicated that the distinct treatment of VLM at tide gauges may be a plausible

explanation of the inconsistent trends before 1990 (Dangendorf et al., 2017; Frederikse et al., 2020). While VLM unrelated to GIA was not accounted for in earlier RSOI reconstructions (Church et al., 2004; Church et al., 2006; Church and White, 2011; Ray and Douglas, 2011; Meyssignac et al., 2012), the Kalman Smoother-based approach accommodates possible VLM unrelated through GIA as local effects in a residual term that does not affect the final estimation of GMSL (Hay et al., 2017). Both approaches are also affected by distinct types of uncertainties related to the representation of regional sea-level variability.

While the RSOI reconstructions often provide good representation of inter-annual variability (Calafat et al., 2014; Dangendorf et al., 2019), they have limitations that are rooted in the assumption that major modes of variability can be robustly determined by the short satellite record and that the corresponding spatial patterns remain stationary in time (Christiansen et al., 2010, Calafat et al., 2014). It is now well established that modes of variability can extend over periods much longer than the satellite record (Chambers et al., 2012; Dangendorf et al., 2014a) and that different contributing factors to sea level, such as

sterodynamic changes or ice-melt, vary in space and time (Marzeion et al., 2012; Frederikse et al., 2020). Consequently, the resulting regional trends over their overlapping period from 1960 to 2007 vary widely between available RSOI reconstructions (Carson et al., 2017). The Kalman Smoother generally overcomes the non-stationary modes in the RSOI-based technique by estimating the individual GRD fingerprints directly from the tide gauge record at each time step and by using sterodynamic priors from the ocean components of climate models. However, the GRD fingerprints co-vary between different source regions

such that, especially in the presence of large sterodynamic variability (Dangendorf et al., 2021), their separation becomes a low signal-to-noise challenge (Hay et al., 2017). This limitation is further exacerbated by the fact that the sterodynamic priors from climate models are not in phase with observations since climate models all start from their own initial conditions (Fasullo et al., 2018). Thus, there is limited skill in reducing the noise for the detection of GRD fingerprints at inter-annual to decadal scales. While this has been demonstrated to have minor effects on the estimation of GMSL, the inability to separate the various

GRD fingerprints propagates into the resulting regional fields and hampers a separation into individual sea-level contributors (Hay et al., 2017).

Here, we introduce a new reconstruction that leverages the strengths of both approaches. Since the original publications of the Kalman Smoother approach (Hay et al., 2013; Hay et al., 2015), independent estimates of historical GRD fingerprints and



corresponding uncertainties have become available (Frederikse et al., 2018; Frederikse et al., 2020, and references therein),
making their estimation from tide gauge records no longer mandatory. Instead, we pre-describe the entire history of
contemporary barystatic GRD fingerprints (glaciers, ice sheets, and terrestrial sources) and their uncertainties with an ensemble
based on Frederikse et al. (2020). We also consider inverse barometer effects (IBE) and prescribe the priors of the spatial
pattern of sterodynamic sea-level change by using an ensemble of RSOI reconstructions from tide gauges after the removal of
GRD, GIA, and IBE estimates. The adjustments have two main advantages: first, the reconstruction of sea level at regional
and global scales is reduced to the sterodynamic component (and associated residual effects that are not properly captured by
the other components), effectively avoiding any co-variance and low signal to noise associated with distinguishing the
individual fingerprints. Second, the approach enables the estimation of the individual contributors or sea-level change as
constrained by tide gauges. As such we derive a novel global and regional mean sea level (MSL) reconstruction at annual
resolution covering the period from 1900 to 2021. We demonstrate that the new reconstruction accurately reconstructs sea
level from satellite altimetry and tide gauges and provides a novel estimate of the sterodynamic component that is independent
of temperature and salinity observations.

## 2 Methodology and Data

### 2.1 Methodology

Hay et al. (2013, 2015, 2017) developed a Kalman Smoother approach that calculates posterior estimates of GMSL and
associated regional sea-level fields from sparse tide gauge records. In the following we follow closely the methodological
description in Hay et al. (2017). The Kalman Smoother is conditioned on a set of different prior sea-level fields that represent
the individual processes contributing to relative sea-level change (sterodynamic sea level, barystatic GRD, and GIA) and can
be divided into three analysis steps: a forward filtering pass, the backward smoother pass, and the multi-model ensemble
component (Hay et al., 2017). The Kalman filter incorporates several equations that can be separated into two groups of time
and measurement update equations (Kalman, 1960). In the time update equations, a prior estimate of sea level at the time step
$k$ is estimated based on the state of the system at time $k$-1. The state vector $\hat{x}_k$ and the associated covariance $\mathbf{P}_k$ are predicted
moving forward in time using the state transition matrix $\varphi$ and the control input parameter $\mathbf{Bu}_t$:

$$\hat{x}_k^- = \varphi\hat{x}_k + \mathbf{Bu}_k \qquad (1)$$

$$\mathbf{P}_k^- = \emptyset\mathbf{P}_{k-1}\varphi^T + \mathbf{Q} \qquad (2)$$

Here $\mathbf{Q}$ is the process noise covariance and the superscript minus sign indicates that the estimates are based on the prior state
of the system from the time update. The state vector $\hat{x}_k$ contains estimated sea level at 516 tide gauge sites (see section 2.2 for
more details on the selection of input data), 74,742 satellite altimetry locations, and an estimate of globally uniform GMSL
change. The latter is the first adjustment to Hay et al. (2015, 2017), where the vector also contained estimates of 21 melt rates
from mountain glaciers and ice sheets. The second adjustment is related to the input control parameters, which comprised
modelled estimates of sea-level contributions from GIA and ocean dynamics in Hay et al. (2015, 2017). Here, we replace the



input control parameters by a combination of model estimates of GIA (Caron et al., 2018), barystatic GRD (Frederikse et al., 2020), IBE, and sterodynamic sea-level change. The sterodynamic component is pre-calculated using an ensemble of RSOI reconstructions (Calafat et al., 2014) that take the 10 leading modes of variability from empirical orthogonal functions estimated from satellite altimetry and fit them to a subset of tide gauge records. Both satellite altimetry and tide gauges are

corrected for the effects of GIA, barystatic GRD, and IBE before performing the RSOI, therefore reducing the RSOI reconstructions solely to the sterodynamic component of sea-level change (and possible residual processes that are not properly captured by the other processes). Note that the RSOI is performed without the spatially uniform zero matrix (that was introduced by Church et al. (2004) to preserve the long-term trend in GMSL), as we are solely interested in reconstructing regional and global variability. Low-frequency changes (including the trend) of global mean steric sea-level change are

estimated in the Kalman smoothing process. The resulting sea-level fields, that we interpret in terms of sterodynamic variability, are then used as input parameters for the Kalman Smoother. The state transition matrix $\varphi$ describes prior beliefs about how sea level rates vary from one time step $k$ to another. It also contains information about the spatial patterns of sea-level change that connect the global mean to any given locality.

During the measurement update, prior estimates of $\widehat{\mathbf{x}}_k$ and $\mathbf{P}_k$ are conditioned upon tide gauge observations stored in $\mathbf{z}_k$ at time

step $t_k$ to identify a linear combination of the prior estimate $\hat{\mathbf{x}}_k^-$ and a weighted difference between observations and predictions $\widehat{\mathbf{H}\mathbf{x}_k^-}$, where $\mathbf{H}$ is a matrix that maps the state vector and covariance into the observation space:

$$\hat{\mathbf{x}}_k = \hat{\mathbf{x}}_k^- + \mathbf{K}_k(\mathbf{z}_k - \mathbf{H}\hat{\mathbf{x}}_k^-) \text{ and} \tag{3}$$

$$\mathbf{P}_k = (\mathbf{I} - \mathbf{K}_k\mathbf{H})\mathbf{P}_k^-. \tag{4}$$

$\mathbf{I}$ is the identity matrix, and $\mathbf{K}_k$ is the Kalman gain matrix defined as follows:

$$\mathbf{K}_k = \mathbf{P}_k^-\mathbf{H}^T(\mathbf{H}\mathbf{P}_k^-\mathbf{H}^T + \mathbf{R})^{-1}, \tag{5}$$

where $\mathbf{R}$ is the observation noise covariance matrix (Kalman, 1960). The observation noise covariance matrix $\mathbf{R}$ is calculated using an expectation-maximization approach, which identifies the covariance for all 516 tide gauge records in the presence of data gaps using an iterative maximum likelihood estimate (Schneider, 2007). To avoid large biases due to data gaps the algorithm is only applied to data after 1955, after which data gaps significantly decrease. Further details on the calculation of

$\mathbf{R}$ can be found in the supplementary information of Hay et al. (2013).

After the forward pass, the filter is run backwards in time and a linear combination of the forward and backwards passes is calculated to ensure that the posterior estimates of the state vector and its covariance over the entire time window (here 1880 to 2021, although we only present results since 1900 to avoid drifts following Hay et al. (2015)) are conditioned upon all observations (Gelb et al., 1974).

The last step comprises the ensemble multi-model component, in which the forward and backwards passes are calculated for all possible prior combinations. Here, we consider 40 different GIA models, 100 GRD realizations, and one IBE product resulting in a total of 4,000 different combinations (i.e., also 4,000 different RSOI realizations) considered in the multi-model component. The likelihood of each combination of models is calculated based on available observations (see Hay et al. (2013) for details), and a weighted sum of results is obtained. It is this weighted sum that represents the final ensemble multi-model



Kalman Smoother GMSL estimate and the associated regional field. As outlined in Hay et al. (2013, 2017), a residual term in the Kalman Smoother allows for modelling local effects at individual tide gauge sites (e.g., vertical land motion unrelated to GIA or possible datum shifts), but this term is not included into the computation of GMSL and the associated regional field. The Kalman Smoother provides a rigorous probabilistic framework for the propagation of observational and model errors. In this current version, uncertainties $\sigma$ are determined by $\mathbf{P}_k$ for each individual tide gauge and the resulting GMSL estimates.

The uncertainty at each of the 74,742 grid points of the regional sea-level fields is determined by the underlying ensembles of the 4,0000 different sea-level contributor fields considering their probability weights plus the spatially uniform uncertainty from the GMSL estimation. Further details on the Kalman Smoother technique can be found in Hay et al. (2013, 2015, 2017). Readers who are interested in further details of the RSOI technique are referred to Calafat et al. (2014).

**2.2 Data**

We use 516 globally distributed annual tide gauge records from the online portal of the Permanent Service for Mean Sea Level (PSMSL) in Liverpool (Holgate et al., 2013) that provide at least 20 years of data and are located south of 67° north (thus ignoring the uncertain records in the Russian Arctic; Hamlington and Thompson, 2015). We also performed an additional visual screening of the records that fulfill the former criteria. Sites that are obviously affected by strong nonlinear VLM due to either earthquakes (e.g., many sites around Japan, or Ko Lak in Thailand after the 2004 Tsunami) or fluid withdrawals (e.g., in Louisiana and Texas) are either entirely discarded, or only portions of the record have been considered (e.g., Manila before the 1960s). All tide gauges with their geographical location and their temporal availability are shown in **Figure 1**.

For GIA, we consider 40 models taken from the ensemble published by Caron et al. (2018), who calculated GIA uncertainties using Bayesian techniques for an ensemble of 128,000 forward models based on varying 1-dimensional Earth structures and ice histories. Thirty of the GIA models were selected based on similar statistics of the large ensemble, that is: they were selected such that their median and standard deviation matches that of the large ensemble (Lambert Caron, *pers. comm.*). We also added the 10 best performing models based on their likelihood inferred from the comparison to Global Navigation Satellite System (GNSS) and relative sea level records in Caron et al. (2018). We consider relative sea level (for tide gauge records) and sea-surface height outputs (for satellite altimetry) from the models.

Instead of calculating melt rates from tide gauge records within the Kalman Smoother framework, we incorporate recently published barystatic GRD estimates from Frederikse et al. (2020). The estimates have been updated to 2020 (Thomas Frederikse, *pers. comm.*); therefore, the data was extrapolated to the year 2021 based on a linear fit to the last 5 years at each grid point (using either a linear or quadratic fit does not significantly affect the extrapolation over one year). The GRD ensemble consists of 100 representative members (again, these were chosen such that they have a similar median and standard deviation as the 5,000 member ensemble from Frederikse et al., 2020) that combine mass changes from glaciers (Parkes and Marzeion, 2018; Zemp et al., 2019), ice sheets (Kjeldsen et al., 2015; Imbie et al., 2018; Bamber et al., 2018; Adhikari et al., 2019; Mouginot et al., 2019) and terrestrial water storage including hydrology (Humphrey & Gudmundsson, 2019), water impoundment in artificial reservoirs (Chao et al., 2008) and groundwater depletion (Döll et al., 2014; Wada et al., 2016).

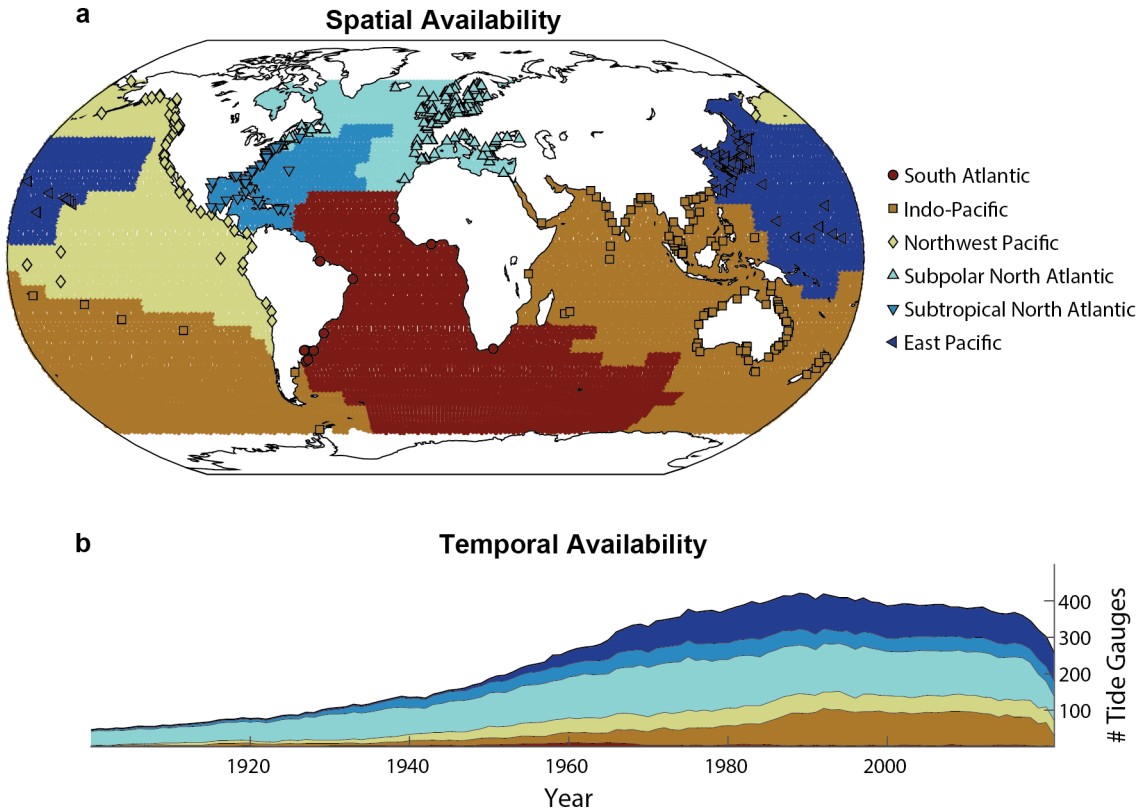

**Figure 1: Global map of tide gauges used in this study and their net availability in time. a Shown are tide gauges as black edged**
**markers with colors representing different regions. The colored shadings on the map represent the six regions used for basin-scale**
**averages described in Thompson and Merrifield (2014). b The corresponding availability as a function of time differentiated for**
**each region.**

IBE variations are derived from sea-level pressure data from atmospheric reanalysis models, specifically the 20[th] century

reanalysis v3 covering the period 1900 to 2015 (Slivinski et al., 2019) and the NCEP-NCAR reanalysis 1 (Kalnay et al., 2018)

from 2016 to 2021. Both reanalyses were interpolated from different global grids onto the 74,742 grid points for the regional

sea-level fields considered in the Kalman Smoother. They are combined after adjusting them to the same mean at each grid

point over the five overlapping years between 2011 and 2015. IBEs are calculated from the combined and interpolated sea-

level pressure fields as follows:

$$\mu^{IBE} = -\frac{P_a - \bar{P}_a}{\rho g} \quad , \quad (6)$$

where $\mu^{IBE}$ is the inverted barometer contribution to sea level, $P_a$ the atmospheric pressure at sea level (the overbar represents

the spatial average over the global ocean surface: ~1013 hPa), $\rho$ is a constant reference ocean density, and $g$ the acceleration

due to gravity (Ponte, 2006).

Satellite altimetry is required for the calculation of the RSOI reconstruction of residual sea level, which interpret as

sterodynamic. Here we use monthly sea-surface height fields from Copernicus Marine Environment Monitoring Service

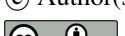



(CMEMS) (Copernicus Climate Change Service, Climate Data Store, 2018), which are available on a global grid of 0.25°
resolution and cover the period 1993 to 2021. To make the satellite data comparable to the relative sea level at tide gauges, we
add the dynamic atmospheric correction that represents IBE variations at timescales longer than 20 days (Ponte and Ray, 2002)
and the deformational component of barystatic GRD (Frederikse et al., 2020). We also correct the satellite record for a globally
uniform drift of about -1.5 mmyr$^{-1}$ from the TOPEX mission before 1999 following WCRP (2018). At higher latitudes, we fill

data gaps due to seasonal sea-ice using a DINEOF approach (Coulsen et al., 2021) if records provide at least 50% data coverage
over the entire period. We then built annual averages based on the gap-filled monthly data. For computational reasons we only
consider every fourth grid point leaving us with a total of 74,742 locations.

## 3 Results

### 3.1 Validation

The GMSL and basin-scale averages (based on basin definitions following Thompson & Merrifield, 2014) of the new
reconstruction of relative sea level and its individual contributors are presented in **Figure 2**. Overlaid are the global and
regional averages from satellite altimetry. The correlations between (detrended) basin-scale averages of the reconstruction and
satellite altimetry observations are all larger than r = 0.89 with maximum values of r = 0.99 in the eastern Pacific (**Tab. 1**).
The high level of agreement between both products also holds at local levels, which is demonstrated by the correlation map in

**Fig, 2a**. The median correlation over the entire oceans worldwide is about r = 0.87 with more than 95% of all grid points
having a correlation coefficient of 0.7 or higher. In general, correlations are highest closer to the coast and in tropical latitudes
of the Pacific and Indian Oceans. The higher correlation along coasts may be related to propagating signals that induce
coherence over large distances and are often linked to large scale modes of variability (Hughes et al., 2019). The high
correlations in tropical latitudes of the Pacific and Indian Ocean is due to the dominant influence of Pacific climate variability

(El Nino Southern Oscillation and Pacific Decadal Oscillation) onto regional sea level worldwide (Hamlington et al., 2013).
Lowest correlations are found, as common in other spatiotemporal reconstructions (Carson et al., 2017; Dangendorf et al.,
2019), along the pathways of major ocean circulation systems such as the Gulf Stream, the Kuroshio, or the Antarctic
Circumpolar region. This is because those areas are characterized by considerable small-scale variability due to the presence
of mesoscale eddies, which reduces, in combination with the sparse sampling of tide gauges, the skill of the reconstruction.

The reconstruction is also meant to capture primarily large-scale patterns by reducing it to the 10 major modes. However, this
eddy-related noise averages out at basin scales, leading to consistently larger correlation coefficients (**Tab. 1**).



Figure 2: **Global and regional reconstructed sea-level changes, their contributing causes, and comparison to observations from satellite altimetry. Shown is the Pearson correlation coefficient between the (detrended) new reconstruction and (detrended) satellite altimetry at each grid point over the world's oceans (a) and the time series of global (b) and regional basin-scale (c-h) sea level and its causes. In b-h the satellite altimetry record is shown for comparison as well. Uncertainties are shown as shadings.**

The good correspondence between the new reconstruction and satellite altimetry is not only limited to the variability but becomes also visible in the linear trend and acceleration maps (including the solid earth component of barystatic GRD and
relative sea-level signals of GIA as determined by the Kalman Smoother, which has been added to satellite altimetry) over their overlapping period from 1993-2021 (**Figure 3**). The spatial pattern correlation between the two resulting maps for trends and acceleration coefficients (both estimated via ordinary least squares) is r = 0.97 and r = 0.92, respectively. There is no systematic bias between observed and reconstructed trends and accelerations, which is reflected in median differences of about 0.04 mmyr$^{-1}$ and -0.01 mmyr$^{-2}$, respectively. 95% of all locations worldwide show differences in trends and accelerations that
are smaller than an absolute value of 0.85 mmyr$^{-1}$ and 0.26 mmyr$^{-2}$, respectively (**Figure 3**). The majority of trends (98.9% of the entire ocean area) and accelerations (96.5% of the entire ocean area) in the residuals (satellite observations minus reconstruction) are also not statistically significant. Again, at basin scales those differences are even smaller (**Figure 2, Tab. 1**).

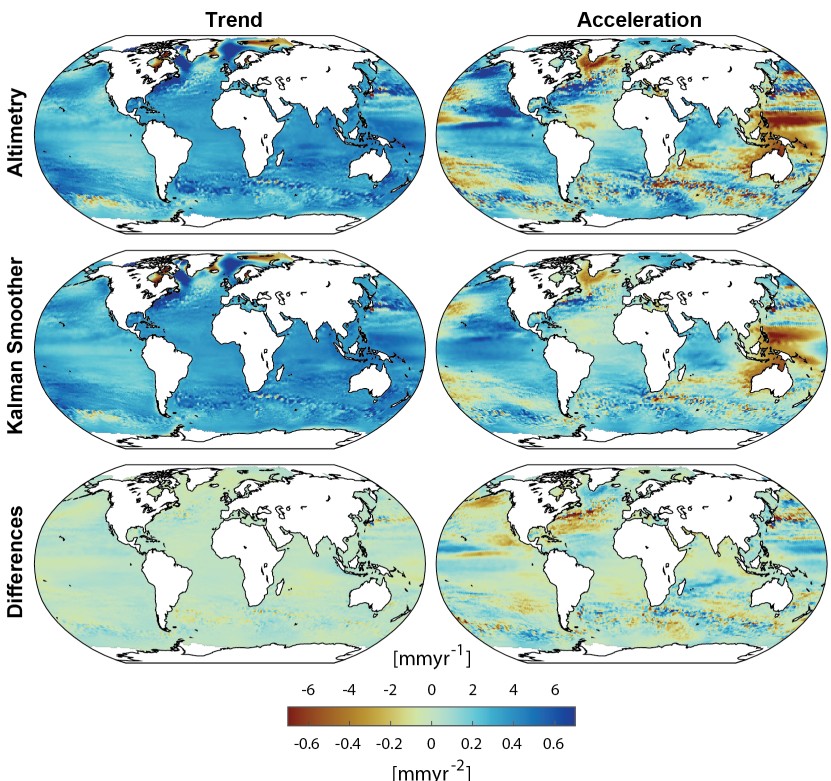

**Figure 3: Trends and acceleration from satellite altimetry and the reconstruction over 1993 to 2022. Trends (left) and acceleration (right) as determined by satellite altimetry (top) and the reconstruction (middle). Also shown are the differences between the two products (satellite altimetry minus reconstruction, bottom). Note that all trend maps show relative sea level trends and thus include the relative sea-level change signal of GIA as determined within the Kalman Smoother reconstruction. The pattern correlation between the two trend maps is r = 0.97 and between the two acceleration maps r = 0.92. The RMS difference is 0.43 mmyr$^{-1}$ and 0.13**
**mmyr$^{-2}$, respectively. Note that the trend and acceleration differences are insignificant over 98.9% and 96.5% of the global ocean, respectively.**



**Table 1: Linear trends in global and basin-scale sea level from the Kalman Smoother and satellite altimetry over 1993 to 2021. All trends have been calculated based on ordinary least squares regression. Trend uncertainties are presented as standard errors assuming that the residuals are autocorrelated and can be reasonably well approximated by an autoregressive process of the order one. They also contain the reconstruction uncertainty. Also shown are the respective correlation coefficients calculated from the detrended global and basin-scale averages.**

| Basin | Linear Trends ± SE [mmyr⁻¹] over 1993 to 2021 | | Acceleration ± SE [mmyr⁻²] over 1993 to 2021 | | Correlation |
|---|---|---|---|---|---|
| | Kalman Smoother | Satellite Altimetry | Kalman Smoother | Satellite Altimetry | |
| South Atlantic | 3.43 ± 0.39 | 3.52 ± 0.21 | 0.13 ± 0.09 | 0.10 ± 0.05 | 0.89 |
| Indo-Pacific | 3.60 ± 0.33 | 3.67 ± 0.19 | 0.07 ± 0.09 | 0.04 ± 0.05 | 0.90 |
| East Pacific | 2.20 ± 1.10 | 2.15 ± 1.09 | 0.33 ± 0.27 | 0.35 ± 0.27 | 0.99 |
| Subpolar North Atlantic | 1.86 ± 0.32 | 1.92 ± 0.35 | -0.04 ± 0.08 | -0.07 ± 0.09 | 0.90 |
| Subtropical North Atlantic | 3.53 ± 0.52 | 3.26 ± 0.62 | 0.23 ± 0.12 | 0.30 ± 0.12 | 0.91 |
| West Pacific | 3.44 ± 0.48 | 3.58 ± 0.44 | -0.01 ± 0.13 | -0.08 ± 0.12 | 0.94 |
| Global | 3.17 ± 0.38 | 3.22 ± 0.21 | 0.12 ± 0.06 | 0.10 ± 0.03 | 0.96 |

An additional independent way of testing the performance of the new reconstruction is by assessing the inferred sterodynamic component against independent estimates of steric height as derived from temperature and salinity observations. Here we make use of various global (WCRP, 2018; Frederikse et al., 2020; Camargo et al., 2022) and basin-scale (Frederikse et al., 2020) estimates, all of them based on ensembles of different products. An important consideration here is that a comparison between steric and sterodynamic height is only straightforward at basin or global scales, as the former does not contain any manometric signals due to ocean circulation changes (Bingham and Hughes, 2012; Dangendorf et al., 2021). At a basin scale, manometric sea-level changes (that manifest in ocean bottom pressure) are thought to be a minor contribution at interannual and longer timescales (Frederikse et al., 2020), while at global scales they entirely cancel out. The results of the comparison are summarized in **Figure 4** and **Tab.2**. Linear trends agree within their respective uncertainties everywhere during the overlapping period from 1958 to 2018 (considering 95% confidence levels, i.e., two times the standard error given in **Tab. 1**) except for the Northwest Pacific and the Subpolar North Atlantic, where the Kalman Smoother reconstruction produces (mean) sterodynamic trends that are 169% (0.49 mmyr⁻¹ difference) and 153% (0.33 mmyr⁻¹) larger than from the steric height fields based on temperature and salinity observations. Although they agree within their uncertainties, trends in the Indo-Pacific region are also overestimated by 154% (0.35 mmyr⁻¹) by the Kalman Smoother reconstruction. There are several possible reasons for that mismatch. The reconstruction in the Northwestern Pacific primarily relies on tide gauges that are situated in areas of high seismic activity (Oelsmann et al., 2023). While we have screened the dataset before inclusion into the reconstruction, we cannot rule out that residual signals propagate into the reconstruction of the spatial portion of the sterodynamic component as determined by the RSOI. Furthermore, both regions and specifically the Subpolar North Atlantic contain large shallow continental shelf seas, in which ocean bottom pressure changes (i.e., manometric sea-level changes) dominate the sterodynamic variability (Landerer et al., 2007). Thus, the larger trends may also indicate an increase in wind-driven manometric changes



onto the shallow shelf that would not be seen in steric height alone. Indeed, large portions of the Northwestern European shelves have experienced positive trends due to shifts in the North Atlantic Oscillation and the corresponding westerly winds

over the second half of the 20th century (Dangendorf et al., 2014b; Gräwe et al., 2019). As a test we calculated the basin average in the Subpolar North Atlantic with and without the Northern European shelf. Without the Northern European Shelf, the trend was ~0.1 mmyr$^{-1}$ smaller than with it, thus explaining a maximum of~30% of the total difference. A similar test in the Northwestern Pacific did not yield any significant trend differences. Finally, we use the outputs from one-dimensional GIA models (Caron et al., 2018) within the Kalman Smoother framework, which may perform reasonably on a global scale but are

characterized by larger uncertainty locally (Thompson et al., 2023). Thus, a fraction of the mismatch in the Subpolar North Atlantic might also arise from uncertainties in GIA models propagating into the sterodynamic reconstruction. Lastly, we note that temperature and salinity-based steric height products are deeply uncertain as well (e.g., MacIntosh et al., 2016; Rietbroek et al., 2016; Cheng et al., 2020) and contain only very little information about the deep ocean below 2000m. Overall, however, the agreement between the two products is therefore reasonable. This is also reflected in correlations, which range from r =

0.57 in the South Atlantic to r = 0.87 in the East Pacific, indicating a moderate to strong agreement in the variability in all basins (**Tab. 2**). Similar conclusions can be drawn for the global mean. The linear trends between the Kalman Smoother reconstruction and different steric height estimates agree within their respective uncertainties (central trend in the Kalman Smoother reconstruction is only 0.17 mmyr$^{-1}$ larger than the ensemble mean from Frederikse et al. (2020)) and the variability is significantly correlated (r = 0.68) over the period 1958 to 2018. Before 1958, the global average from Frederikse et al. (2020)

is based on only one available reconstruction from Zanna et al. (2019). Both show some similarities in the first decades of the 20th century, although the steric height estimate from Zanna et al. (2019) exhibits slightly stronger and more enhanced warming in the 1930s than the Kalman Smoother estimate from tide gauges (**Figure 4a**). Again, those differences are largely within the respective error bars.

**Tab. 2 | Performance metrics of the Kalman Smoother reconstruction. Shown are the linear correlation coefficients between**
**detrended basin-scale and global averages of sea level from the Kalman Smoother reconstruction and satellite altimetry as well as the sterodynamic sea level (SDSL) component from the Kalman Smoother reconstruction and steric height from Frederikse et al. (2020) (left). Also shown are the linear trends from the latter two over their overlapping period from 1958-2018. Trend uncertainties are shown as standard errors assuming that the residuals are autocorrelated and can be reasonably well approximated by an autoregressive process of the order one. They also contain the reconstruction uncertainty.**

| Basin | Correlation | | Linear Trends ± SE [mmyr$^{-1}$] | |
|---|---|---|---|---|
| | Kalman Smoother vs. Satellite Altimetry (1993-2021) | SDSL Kalman Smoother vs. Steric Height (1958-2018) | SDSL Kalman Smoother (1958-2018) | Steric Height Frederikse (1958-2018) |
| South Atlantic | 0.89 | 0.57 | 0.86 ± 0.14 | 0.88 ± 0.10 |
| Indo-Pacific | 0.90 | 0.79 | 0.99 ± 0.16 | 0.64 ± 0.22 |
| East Pacific | 0.99 | 0.87 | 0.22 ± 0.23 | 0.47 ± 0.20 |
| Subpolar North Atlantic | 0.90 | 0.61 | 0.95 ± 0.13 | 0.62 ± 0.13 |
| Subtropical North Atlantic | 0.91 | 0.61 | 1.36 ± 0.21 | 1.29 ± 0.20 |
| West Pacific | 0.94 | 0.79 | 1.20 ± 0.18 | 0.71 ± 0.19 |
| Global | 0.96 | 0.68 | 0.87 ± 0.10 | 0.71 ± 0.11 |


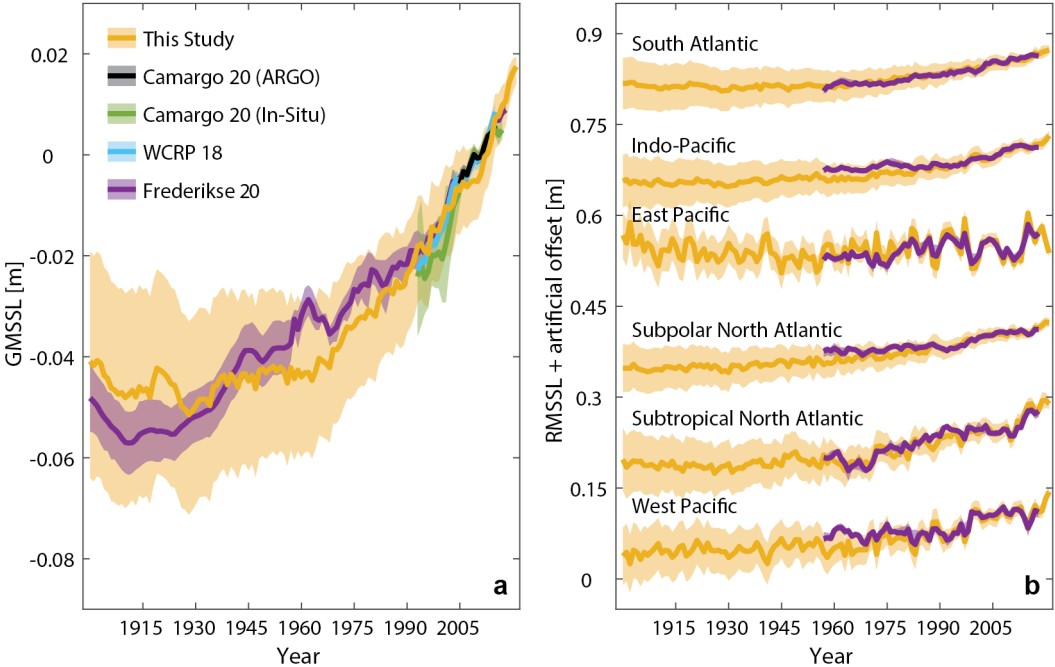

**Figure 4: Comparison between observed steric and reconstructed sterodynamic global and regional mean sea level. a Global mean (thermos-)steric sea level from different observational products based on temperature observations versus the new estimate from the reconstruction. b Basin-scale regional averages of steric height from observational products (Frederikse et al., 2020) and the new sterodynamic estimate from the reconstruction. Uncertainties are shown as shadings.**

Next, we look at individual tide gauge sites and compare them with the time series of the nearest neighbor grid point from the weighted fields of the Kalman Smoother reconstruction and satellite altimetry. **Figure 5 a-g** shows seven worldwide distributed sites: San Francisco (United States), Brest (France), Nagasaki (Japan), Honolulu (Hawaii), La Libertad (Ecuador), Fremantle (Australia), and Chuuk, Moen Island (Micronesia). We note that the nearest neighbour time series from the weighted fields of the Kalman Smoother do not contain the residual term from the Kalman Smoother and are thus purely the result of the known modelled processes (i.e., the sum of sterodynamic, barystatic GRD, IBE, and GIA) at each location (Hay et al., 2017). At all seven locations the nearby reconstruction provides a good representation of the interannual variability, which is reflected in strong correlation coefficients that range from r = 0.75 in Brest to r = 0.91 in Chuuk, Moen Island. Linear trends over their overlapping periods agree within their respective uncertainties and trend differences are everywhere smaller than 0.6 mmyr⁻¹. We note that the agreement between satellite altimetry and the Kalman Smoother reconstruction is consistently better than with the tide gauge records. This is because the Kalman Smoother best represents large-scale processes (with small-scale processes being absorbed by the residual process), but it also reflects dependence of the reconstruction on the quality of satellite altimetry in the coastal zone. The latter may miss some fundamental processes that take place very close to the shore (e.g., Cazenave et al., 2022) such as coastally trapped signals from wind (Calafat and Chambers., 2013) and river discharge (Piecuch



et al., 2018) or nonlinear VLM rates (Oelsmann et al., 2023). This limitation is evident by comparing satellite altimetry observations and the reconstruction during individual peaks such as the 1997/1998 El Nino with the significantly smaller amplitude signal in the tide gauge record of San Francisco (**Figure 5a**).

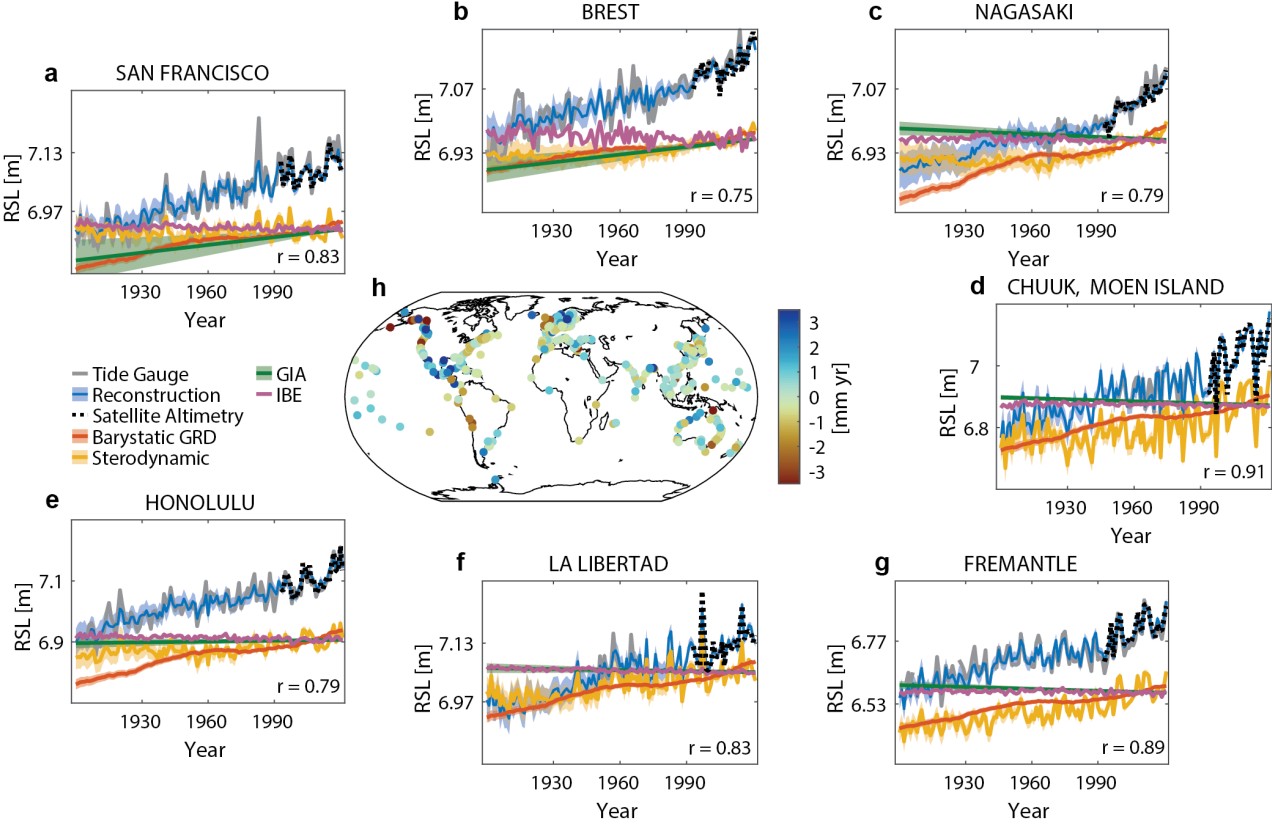

**Figure 5: Estimates of relative mean sea level and individual contributing factors at tide gauge locations and related biases. Shown**
**are tide gauge and satellite altimetry records (with GIA added from the reconstruction) at seven selected locations (a-g) together with the new reconstruction and the estimates of individual contributing factors. Uncertainties are shown as shadings. Correlations between the reconstruction and the tide gauge are shown in the lower right of each panel. h Differences between linear trends estimated from tide gauge records and the new reconstruction over the overlapping period between 1900 and 2022.**

**Fig, 5h** shows the linear trend differences between tide gauge records and the nearest neighbour time series of the weighted
fields of the Kalman Smoother reconstructions at all 516 sites during the overlapping period at each location. The median difference over all sites is 0.1 mmyr⁻¹ with a standard deviation of 1.8 mmyr⁻¹. This suggests that while there is only a minor systematic bias (0.1 mmyr⁻¹) between tide gauge observations and the reconstruction, there are many sites where the observed trends cannot solely be explained by the sum of individual components. 64% of all sites show trend differences below 1 mmyr⁻¹, while a trend difference of about 0.5 mmyr⁻¹ or less is observed at 33% of all sites. The largest and sometimes systematic
differences occur in GIA hotspots of Scandinavia and the North American west coast, in parts of the Gulf of Mexico, and the Pacific coast of South America. There are several possible explanations for those larger differences: First, our process estimates





may have larger uncertainties than implied in the pre-described ensembles including those resulting from limitations in one-dimensional GIA modelling (Thompson et al., 2023). Second, there may be processes that affect tide gauges at the coast but that are not explicitly included as priors in the Kalman Smoother fields. While those can, in principle, be captured by the

residual term at individual tide gauge sites, they are not part of the global and regional sea level reconstructions (Hay et al., 2015; Hay et al., 2017).

The good agreement between the reconstruction and both satellite altimetry and steric height at basin scales, however, suggests that both the sum and individual components are generally well captured by the reconstruction. Indeed, when comparing to the 516 tide gauge records over the satellite period from 1993 to 2021, trend differences are in the order of 0.2±2.17 mmyr[-1].

The trend differences are -0.06±0.42 mmyr[-1] when considering satellite altimetry at the same locations. Thus, there may indeed be other processes such as residual VLM unrelated to GIA that are captured by tide gauges but not included within the weighted fields of the Kalman Smoother reconstruction. To test this hypothesis, we downloaded VLM rate estimates from GNSS as provided by SONEL (Gravelle et al., 2023), which are available for 601 locations worldwide that provide a minimum observation length of seven years. Not all GNSS locations align with the tide gauge sites in our assessment. Accordingly, we

only consider VLM estimates from locations that have a maximum distance of eight kilometres (this is the number below which results remain essentially unchanged) to the tide gauge records, and where at least 75% of the period since 1993 is covered by the tide gauge record. This results in 90 sites worldwide. At those 90 locations we replace the weighted GIA estimate from the Kalman Smoother reconstruction by the VLM estimate. To avoid double-counting of the VLM effects related to contemporary GRD, we removed the linear rate estimate of its deformational component since 1993 from GNSS estimates.

We also added the GIA sea-surface height component from the weighted Kalman Smoother GIA fields to the GNSS estimates (Tamisiea, 2011). The trend comparison is performed over the period 1993 to 2021 to ensure that the time window relative to GNSS observations is comparable. The trend differences between observed tide gauge records, the nearby weighted Kalman Smoother fields, and the nearby weighted Kalman Smoother fields with GNSS-based VLM are shown in **Figure 6** as spatial maps (**Figure 6a,b**) and empirical distributions (**Figure 6c**). The trend differences for the 90 selected sites, 0.27±1.67 mmyr[-]

[1], are not significantly different to those obtained at all 516 sites (0.1±1.8 mmyr[-1]). However, replacing the GIA crustal rates with the GNSS-based VLM significantly reduces the trend differences to -0.14±1.00 mmyr[-1] (i.e., a reduction by 40% in the standard error). Applying the same test to the entire period over which tide gauges provide data results in very similar results (reduction from 1.5 to 1 mmyr[-1] standard error at 119 sites). The reduction is most prominent at the locations of former ice sheets in Scandinavia and North America, but also visible at sites that are known to be affected by residual VLM such as along

the Japanese coastline. Thus, a fraction of the observed differences to tide gauge observations are likely related to unaccounted VLM (in addition to shorter-term variations, for instance, associated with near-coastal processes that are neither captured by satellite altimetry nor the reconstruction). This result also indicates that the Kalman Smoother framework is robust against potential biases through these residual processes. Future work might assess the suitability of the framework for the assessing of nonlinear VLM at individual tide gauge sites (Kopp et al., 2014; Dangendorf et al., 2021).



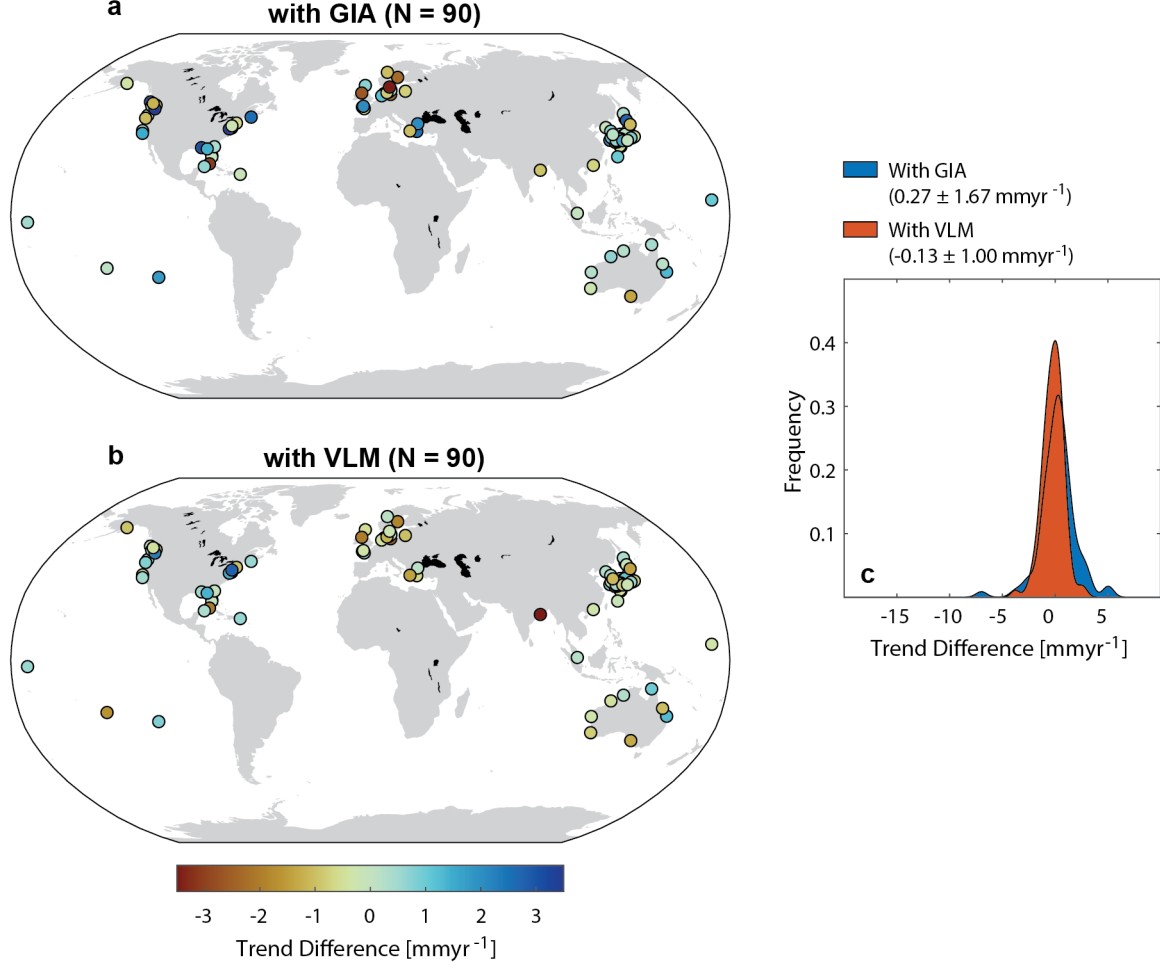

**Figure 6: Linear trend biases at tide gauge sites and the role of VLM.** Shown are the linear trend differences between tide gauge records and the new reconstruction over the overlapping period between 1993 and 2021 considering reconstructed GIA as VLM (a) and an alternative estimate where the reconstruction is combined with observed VLM from GNSS (corrected for VLM from contemporary GRD and with sea-surface height changes due to GIA added) (b). The corresponding Kernal density estimates of trend differences between tide gauge records and the reconstruction over their overlapping period from 1993 to 2021 is shown in c.

**3.2 Trends and variability in GMSL**

After demonstrating the performance of the new reconstruction, we now turn our attention to the modelled changes over the 20th century. Again, the respective global and basin-scale time series of sea level and individual contributors are shown in **Figure 2** with their respective linear trends summarized in **Tab. 3**. In the new reconstruction, GMSL has been increasing by a linear rate of about 1.5±0.19 mmyr$^{-1}$ since 1900. This is lower than the reported 1.7 mmyr$^{-1}$ by Church and White (2011) and the 2.10 mmyr$^{-1}$ by Jevrejeva et al. (2014) (both for shorter periods), but consistent with more recent reconstructions that consider the spatial patterns of individual sea-level processes as prior estimates in the reconstruction process (Hay et al., 2015; Dangendorf et al., 2017; Dangendorf et al., 2019; Frederikse et al., 2020).





**Table 3: Linear trends in global and regional sea level and individual contributors over 1900 to 2021. All trends have been calculated based on ordinary least squares regression. Trend uncertainties are presented as standard errors assuming that the residuals are autocorrelated and can be reasonably well approximated by an autoregressive process of the order one. They also contain the reconstruction uncertainty.**

| Basin | Linear Trends ± SE [mmyr$^{-1}$] | | | |
| --- | --- | --- | --- | --- |
| | Relative Sea Level | Sterodynamic Sea Level | Barystatic GRD | Glacial Isostatic Adjustment |
| South Atlantic | 1.70 ± 0.12 | 0.41 ± 0.15 | 1.28 ± 0.35 | -0.05 ± 0.02 |
| Indo-Pacific | 1.47 ± 0.14 | 0.51 ± 0.13 | 1.13 ± 0.29 | -0.18 ± 0.02 |
| East Pacific | 1.20 ± 0.10 | 0.08 ± 0.09 | 1.13 ± 0.28 | 0.10 ± 0.04 |
| Subpolar North Atlantic | 1.12 ± 0.08 | 0.57 ± 0.08 | 0.32 ± 0.07 | 0.28 ± 0.13 |
| Subtropical North Atlantic | 2.20 ± 0.12 | 0.71 ± 0.14 | 0.97 ± 0.19 | 0.58 ± 0.15 |
| West Pacific | 1.71 ± 0.10 | 0.60 ± 0.09 | 1.21 ± 0.31 | -0.06 ± 0.04 |
| Global | 1.50 ± 0.20 | 0.44 ± 0.24 | 1.10 ± 0.27 | -0.02 ± 0.01 |

It is well known that the reconstructions generally agree well since the 1970s, while they diverge before (Oppenheimer et al., 2019; Palmer et al., 2021). Frederikse et al. (2020) estimated GMSL, but also compared it to the sum of independently measured/modelled contributions. While the long-term trends over 1900 to 2018 of both agreed within their respective uncertainties (1.56±0.2 mmyr$^{-1}$ for GMSL compared to 1.52±0.2 mmyr$^{-1}$ for the sum of individual contributions), some differences appear between both curves between the 1920s and early 1950s. All former GMSL reconstructions, including the Frederikse et al. (2020) curve, show enhanced rates peaking in the 1940s and 1950s, while the sum of individual contributions peaks in the 1930s. This is illustrated in **Figure 7b**, in which rates of GMSL rise were calculated using a Singular System Analysis with an embedding dimension of 15 (thereby smoothing the time series to time scales longer than 30 years; Moore et al., 2005) (error bars consider the reconstruction uncertainty). Interestingly, in the new reconstruction, rates peak earlier and closely resemble the sum of individual components from Frederikse et al. (2020). We attribute this behaviour to the inclusion of barystatic estimates in the new Kalman Smoother framework, which changes the regional sea-level fields near major melt-sources and then propagates these into the estimation of residual sterodynamic fields. This becomes visible when comparing the differences between the regional sea-level rates from this study minus those from Dangendorf et al. (2019) over the period from 1930 to 1939 (**Figure 8**). Notable differences occur in the vicinity of major melt sources (i.e., Greenland, Svalbard, and Alaska), confirming that the original Kalman Smoother framework is not able to properly assign the sea-level signal during this period to its individual barystatic components given the available number of tide gauges and their location in the far-field of melt sources (Hay et al., 2017). Here, this problem is overcome, by pre-describing the melt-signals in the initial prior estimates. We also find significant differences in the Pacific region, indicating that the inclusion of the barystatic estimates impacts the estimation of the spatial pattern of sterodynamic sea-level change. In Dangendorf et al. (2019) the RSOI was



applied to the full sea-level signal and therefore also included potential short-term effects of barystatic GRD processes, while here the RSOI is solely applied to residual signals after accounting for all other budget components.

We note, however, that the overall evolution of multidecadal rates in GMSL remains generally consistent with moderate rates at the beginning of the 20[th] century, enhanced rates in the 1930s, a slowdown with a low in the 1960s, and a persistent increase in the rates of rise thereafter (Dangendorf et al., 2019). Consequently, central estimates of multidecadal trends have exceeded 4 mmyr$^{-1}$ since 2019 for the first time in the observational history, leading to an overall acceleration of about 0.08±0.04 mmyr$^{-2}$ since 1970 (**Figure 7b**). Over the shorter altimetry period since 1993, acceleration coefficients are slightly larger, but

consistent within the error, with values of 0.1±0.06 mmyr$^{-2}$ and 0.12±0.1 mmyr$^{-2}$ for satellite altimetry and the Kalman Smoother reconstruction, respectively.

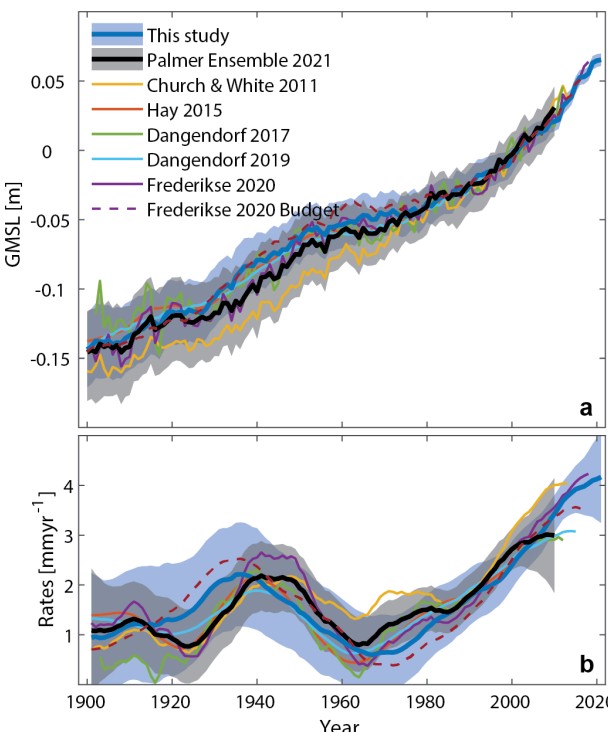

**Figure 7: GMSL and nonlinear rates in comparison to former reconstructions. a Shown are GMSL time series from this study (1900 to 2022) in comparison to selected former reconstructions. All time series have been adjusted to a common mean of zero over the**
**overlapping period from 1993 to 2010. b The corresponding nonlinear trends as determined by a Singular Spectrum Analysis with an embedding dimension of 15, which extracts signals that are representative of timescales longer than 30 years. Uncertainties are shown as shadings and represent 95% confidence intervals.**

Steric and barystatic sea level have contributed about one third and two thirds to the increase in GMSL since 1900, respectively

(**Tab. 3**). For the steric contribution rates of sea-level rise have been consistently increasing from slightly negative rates in

1900 to rates close to 2 mmyr$^{-1}$ in 2021. The negative rates in the early 20[th] century are consistent with the steric reconstruction from Zanna et al. (2019) and global atmospheric mean surface temperature reconstructions (Lenssen et al., 2019), indicating



that both the ocean and the atmosphere were globally in a cooling phase during that period. Barystatic contributions to sea level have been more variable than the steric height with peak rates in the 1930s and since the early 2000s and near-zero rates in the late 1960s and early 1970s (**Figure 9**). This period of low rates has previously been attributed to the increased activity

in dam building at that time (Frederikse et al., 2020), preventing freshwater from flowing into the ocean. The increase in the rates since the early 1970s was initially driven by the steric contribution and later by the barystatic component of sea-level rise. Overall, barystatic sea-level change is the dominant factor behind the acceleration since 1970 (**Figure 10**) although this perspective would change if considering individual sources of barystatic sea level (i.e., glaciers, Greenland icesheet, Antarctic icesheet, hydrology); in that case steric height would be the single most important driver of acceleration since 1970, consistent

with Dangendorf et al. (2019).

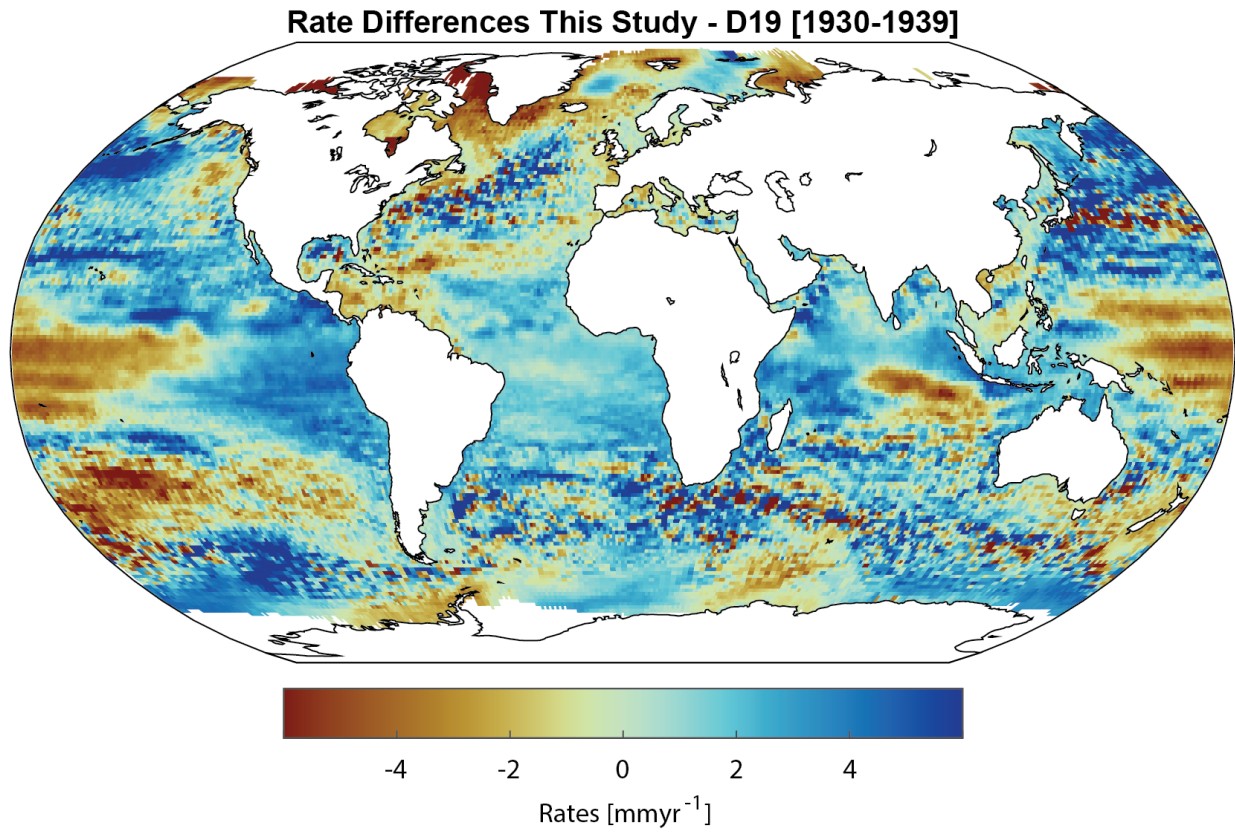

**Figure 8: Rate differences between this study and the regional sea-level fields from Dangendorf et al. (2019; D19) over the period 1930 to 1939. Differences have been calculated between GIA-corrected fields and thus primarily reflect differences between GRD, sterodynamic sea level, and IBE effects.**






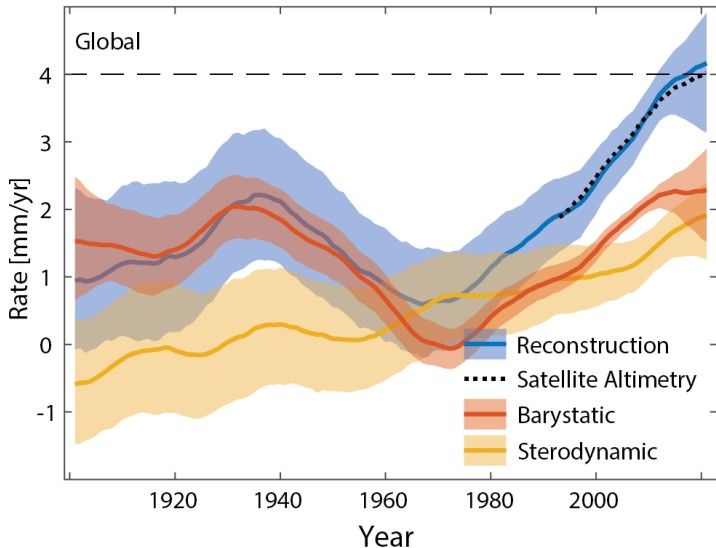

**Figure 9: Rates of GMSL rise and its causes. Shown are the nonlinear rates of GMSL rise and its individual contributors as determined from the Kalman Smoother framework and satellite altimetry. Rates have been calculated with a Singular System Analysis using an embedding dimension of 15 (equal to a cutoff period of 30 years).**





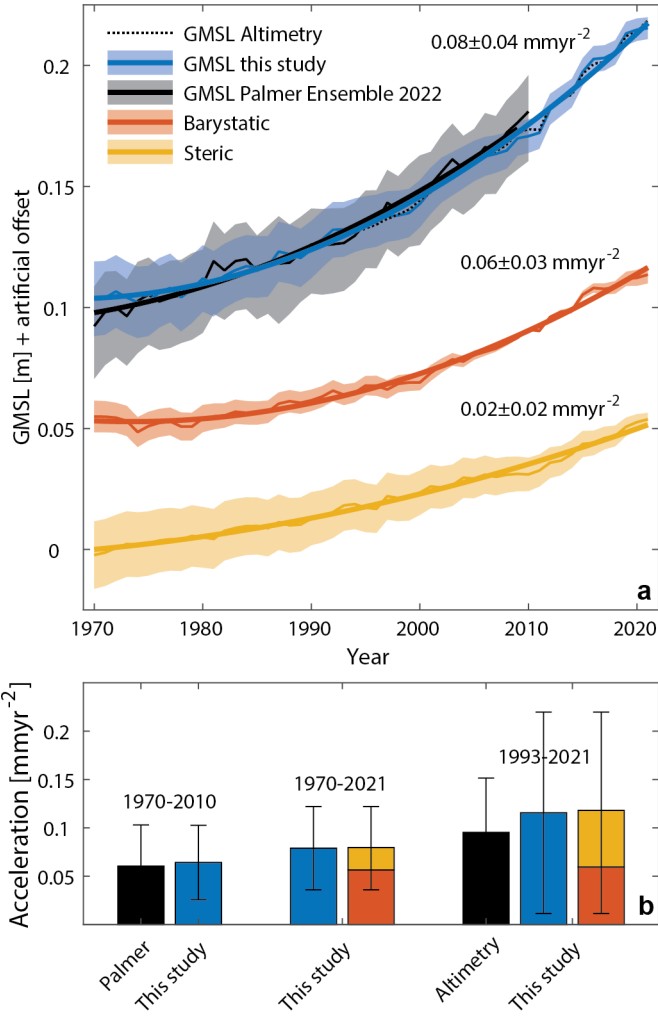

**Figure 10: GMSL acceleration since 1970 and individual contributors. a** Time series of GMSL and steric and barystatic contributions since 1970. Also shown for comparison is the GMSL ensemble estimate based on previous reconstructions as assembled by Palmer et al. (2022). Thin lines show the annual reconstructions, thick lines represent a quadratic fit. **b** Quadratic coefficients as estimated over different periods and individual contributors.

### 3.3 Trends and variability in regional sea level

At basin scales, the trends as well as the relative contributions of individual components can vary significantly (**Tab. 3**). Most notably, in the Subtropical North Atlantic relative sea level has increased 146% faster than GMSL leading to rates of about $2.2\pm0.11$ mmyr$^{-1}$. While barystatic GRD contributions have been slightly below the global average due to their proximity to Northern Hemispheric melt sources, sterodynamic and GIA contributions have been rising at faster rates than anywhere else. This is qualitatively consistent with the results from Frederikse et al. (2020) for the shorter period since 1958 and reflects, in terms of the sterodynamic component, the above average increase in ocean heat content in the Subtropical Gyre (Cheng et al.,



2022). The lowest trends over the 20th century occurred in the Subpolar North Atlantic with rates of about 1.12±0.07 mmyr-1. These lower trends are foremost a result of the reduced contributions of barystatic GRD, which has been increasing at a pace 71% less than the global mean (0.32±0.05 mmyr-1 compared to 1.1±0.27 mmyr-1).

**Figure 11: Linear trends in reconstructed sea level and the individual contributing factors over 1900 to 2021.** Shown are the linear trend maps for relative sea level (a), sterodynamic sea level (b), effects of Gravitation, Rotation, and Deformation (GRD, c), Glacial Isostatic Adjustment (GIA, d), and Inverse Barometer Effects (IBE, e). In a, linear trends are presented for all contributors averaged over 5 degrees latitude bands.





An important feature of the new reconstruction is that it provides not only global and basin-scale estimates but also local sea-level changes including estimates of the individual contributors (**Figure 5**). This is a major advance relative to earlier studies particularly with respect to the representation of dynamic redistribution processes within the sterodynamic component. Previous studies of sea-level budgets based on conventional steric height products from temperature and salinity profiles

pointed to the difficulty of properly estimating the manometric sea-level contribution in the coastal zone (Cabanes et al., 2001; Miller and Douglas, 2004; Frederikse et al., 2018; Dangendorf et al., 2021). In contrast, here the sterodynamic signal is entirely determined by coastal signals as measured by tide gauges. This enables us to perform more sophisticated analyses of past trends and accelerations at a global level with focus on individual budget contributions.

**Figure 11** shows linear trend maps for relative sea level and individual sterodynamic, barystatic GRD, GIA, and IBE

contributions since 1900 including five-degree latitudinal averages of the corresponding trends. Linear trends in the reconstruction range from -4.2 mmyr$^{-1}$ to 11.7 mmyr$^{-1}$ for relative sea levels (**Figure 11a**). Negative trends are primarily located in centers of postglacial uplift where former ice sheets were located (Scandinavia and North America) and are therefore by the GIA signal (**Figure 11d**). The largest positive trends appear in eddy-rich regions near western boundary currents such as the Kuroshio, the Gulf Stream, and the Agulhas system and are related to changes in ocean circulation (i.e., the sterodynamic

component, **Figure 11b**). Those changes are more localized and can reverse sign over a few 100s of kilometers and are generally very uncertain due to the paucity of tide gauge records in those regions. Larger geographical areas characterized by strong positive trends are also found along the U.S. East and Northwest coasts, both of which are related to GIA and specifically postglacial subsidence of peripheral bulges (**Figure 11d**). There is also a strong trend gradient in sea level across the Central Pacific (negative to neutral trends in the east, strong positive trends in the west that also persist into portions of the Indian

Ocean) possibly caused by wind- (Merrifield, 2011) and buoyancy-driven (Piecuch and Ponte, 2012) sterodynamic processes linked to Pacific decadal climate variability (**Figure 11b**). For instance, the Southern Oscillation Index, measuring sea-level pressure differences between Darwin and Tahiti, indicates a weakening over the 20$^{th}$ century (Power & Kociuba, 2010). Similarly visible is the meridional gradient in the South Pacific discussed in greater detail, albeit for shorter time periods, in Volkov et al. (2017) and Dangendorf et al. (2019). Both studies found a link between sea level in this region and wind-forcing

linked changes in the intensity and position of Southern Hemispheric Westerlies. However, more research is warranted to clarify the role of these processes on the centennial timescales as represented here by the linear trends. Barystatic GRD trends are spatially smoother (**Figure 11c**) with strong spatial gradients only in the vicinity of major 20$^{th}$ century melt-sources. Latitudinal averages (**Figure 11a**) indicate a general tendency towards larger trends in lower latitudes with local maxima around ~40° North and South. This pattern is largely dominated by the barystatic GRD terms and results from the fact that

most major melt-sources in the 20$^{th}$ century were in high (northern) latitudes. The local maxima result from additional around ~40 degrees North and South are coming from the sterodynamic contribution and reflect gyre dynamics. IBE contributions are minor everywhere except at latitudes south of 50°S, where trends of up to 0.8 mmyr$^{-1}$ are visible. These trends stem from atmospheric pressure changes linked to a hemispheric-scale intensification and poleward shift of the westerlies in this area

(Swart et al., 2015), although we caution that that the IBE here is based on only one atmospheric reanalysis dataset.

Atmospheric reanalyses are generally known for their uncertainty in this region (Piecuch et al., 2016).

**Figure 12: Acceleration coefficients for reconstructed sea level and the individual contributing factors over 1970 to 2021. Shown are the acceleration coefficient maps for relative sea level (a), sterodynamic sea level (b), effects of Gravitation, Rotation, and Deformation (GRD, c), Glacial Isostatic Adjustment (GIA, d), and Inverse Barometer Effects (IBE, e). In a, acceleration coefficients are presented for all contributors averaged over 5 degrees latitude bands.**



Next, we turn our attention to the spatial acceleration pattern since 1970 (**Figure 12**). The choice of the period is based on the persistence of the acceleration visible in GMSL rise since 1970 (**Figure 10**). There is a large spatial variability in acceleration coefficients ranging from negative values (i.e., deceleration) as low as -0.8 mmyr$^{-2}$ to positive accelerations as large as 1.17 mmyr$^{-2}$. Thus, regionally the acceleration coefficients, even over this 52-year period since 1970, can vary by an order of magnitude relative to the global mean value (0.08 mmyr$^{-2}$). This indicates that acceleration coefficients are very sensitive to regional redistribution processes and natural climate variability, hampering a robust and timely detection of more persistent forced long-term signals (Haigh et al., 2014; Dangendorf et al., 2014; Hamlington et al, 2022). Decelerations in the rate of sea-level rise primarily occur as a result of barystatic GRD effects in the vicinity of major melt-sources, particularly around Greenland (**Figure 12a,c**). There is also one larger spot of pronounced deceleration coefficients in the South Pacific Ocean south of 50°S. Previous studies have indicated that this deceleration is related to an increased amplitude and localized northward shift of Southern Hemispheric Westerlies in the South Pacific (Volkov et al., 2017; Dangendorf et al., 2019). As a result, upper ocean warm water has moved northwestward leading to a deceleration of the sterodynamic signal southward (**Figure 12b**) and, at the same time, the initiation of an acceleration hotspot in the central Subtropical South Pacific. Note that IBE contributions have slightly compensated these sterodynamic effects due to changes in sea-level pressure (**Figure 12e**). The acceleration in the tropical South Pacific further extends into both the equatorial Western Pacific and large parts of the tropical Indian Ocean making it the largest continuous area affected by accelerating sea levels over this period worldwide. This trend is also supported by the latitudinal averages in **Figure 12a**, thus bolstering results from earlier studies based on different techniques (Merrifield et al., 2009) or earlier versions of the Kalman Smoother using different priors (Dangendorf et al., 2019) that indicated an important role of tropical regions and the Southern Hemisphere in driving the recent acceleration in GMSL. Again, barystatic GRD effects (**Figure 12c**) are spatially smoother than their sterodynamic counterparts (**Figure 12b**), and they contribute coherently positively to the acceleration, making it the most dominant contribution in latitudinal averages south of 30°N (**Figure 12a**).

To better place some of the observed decelerations/accelerations into a historical context, we select five case-study sites in areas of strong deceleration/acceleration (the purple boxes in **Figure 13a**). Around Greenland relative sea levels have been alternating around zero since 1900 with positive rates peaking in the 1920s and late 1980s and negative lows in the 1930s and the 2010s (**Figure 13b**). The transition from increasing rates in the 1970s and 1980s to negative rates in the 2000s has led to a deceleration in this area, which is largely driven by barystatic GRD effects (Coulsen et al., 2022). However, we note that the low rates in the 2000s are still slightly higher than the lows in the 1930s and are thus not unprecedented. Qualitatively consistent with Spada et al. (2014), sterodynamic sea level has risen steadily in this area with an average rate of ~0.6 mmyr$^{-1}$. The largest uncertainties in the total rates are related to the GIA contributions, likely due to ice histories and Earth rheology assumptions in modelling (Spada et al., 2014). In the South Pacific, sea levels have been decelerating over this period as well (**Figure 13c**), but the decomposition into individual components is very different compared to the Greenland example. Here, the sterodynamic component has been the dominant source of multidecadal variability in the rates. Over the entire period, sterodynamic sea level has been decreasing by an average rate of ~1 mmyr$^{-1}$. Minima in the signal occurred in the 1940s and





the late 2000s, but they have been partially counterbalanced by barystatic GRD and IBE contributions. In the Gulf Stream (**Figure 13d**) and Kuroshio regions (**Figure 13e**), rates have been rising faster than elsewhere with average rates of approximately 2.7 mmyr$^{-1}$ and 1.9 mmyr$^{-1}$, respectively. This rise is clearly dominated by sterodynamic contributions. In the larger Kuroshio region (**Figure 13e**), rates have consistently increased from 1 mmyr$^{-1}$ in 1980 to 5.8 mmyr$^{-1}$ in the most recent

years, while in the larger Gulf Stream region the increase only began in the late 1990s from 3.1 mmyr$^{-1}$ in 1999 to 7 mmyr$^{-1}$ in 2021. In both regions accelerations have also been reported for the adjacent coastlines of Japan (Usui and Ogawa, 2022) and the United States Southeast (Dangendorf et al., 2023; Yin et al., 2023; Steinberg et al., 2024) with rates exceeding 10 mmyr$^{-1}$ locally. In both regions, sterodynamic variability has been the most dominant contributor to variations in the rates, while barystatic GRD contributed a smoother but consistently positive signal over most of the 20$^{th}$ century. Due to the geographic

proximity to the location of the Laurentide ice sheet, GIA adds more uncertainty to the total rates of relative sea level in the Gulf Stream region than in the Kuroshio region (**Figure 13d,e**). In the Tropical Western Pacific, sea levels have also been rising faster (2 mmyr$^{-1}$) than GMSL (1.6 mmyr$^{-1}$) (**Figure 13f**). High rates (close to 5 mmyr$^{-1}$) occurred in the early 1900s and the late 1990s. Both peaks were dominated by the sterodynamic component and were coeval with minima in the Southern Oscillation Index (representing sea-level pressure differences between Tahiti and Darwin). While sterodynamic rates slightly

decreased after the peak in 1999, rates in relative sea level plateaued due to increasing contributions from barystatic GRD. Both GIA and IBE contributions play a minor role in these areas. In general, we find that sterodynamic mostly dominates nonlinear rates locally at time scales up to a few decades, while barystatic contributions become more important at larger spatial and temporal scales.

Overall, these results underscore the importance of embedding recent accelerations into a historical context. The new Kalman

Smoother reconstruction provides interesting new insights into the causes of sea-level changes at a near-global coverage that complements tide gauge and satellite observations and can motivate more detailed assessments in case-study regions that are beyond the scope of this study.

## 4 Data and Code Availability

We provide access to the full sea-level reconstruction including all contributing processes and related uncertainties. The data

is archived as MATLAB and netcdf files and can be accessed via zenodo under the following link: https://doi.org/10.5281/zenodo.10621070 (Dangendorf, 2024). Data is provided as a spatiotemporal field for each individual component with 74,742 grid points, a separate uncertainty field, and the associated global- and basin-scale averages and their related uncertainties. Codes for the analysis of the sea-level fields and the production of the figures are also provided. The code for the Kalman Smoother reconstruction is currently still prepared for publication but available from the main author

upon request. We intend full publication of the Kalman Smoother approach with the next version of the reconstruction with cleaned-up and annotated codes.



**Figure 13: Acceleration since 1970 and nonlinear rates in selected hotspot regions around the world. a Map of acceleration coefficients determined over the period from 1970 to 2022 (same as Fig. 6) together with selected areas (purple boxes), for which average rates of sea-level rise are shown in b-f. Rates of nonlinear relative sea-level rise and individual contributing factors as determined by the new reconstruction are shown for Greenland (b), the South Pacific (c), the Gulf Stream extension region (d), The Kuroshio extension region (e), and the Tropical Western Pacific (f). Nonlinear rates have been determined with a Singular Spectrum Analysis with an embedding dimension of 15, producing trends that are representative for timescales longer than 30 years. Uncertainties are shown as shadings and represent 95% confidence intervals.**





## 5 Discussion and Conclusions

Here we have introduced a new Kalman Smoother based estimate of global and regional sea-level changes and their causes over the period from 1900 to 2021. Innovations compared to earlier versions (Hay et al., 2015; Dangendorf et al., 2019) are our adoption of adjusted estimates of GIA, observation-based and reconstructed barystatic GRD (as a combination of
contributions of glaciers, ice sheets, hydrology, and terrestrial water sources), IBE contributions, and (reconstructed) sterodynamic variability. As a result, the new reconstruction moves beyond global and regional sea-level estimates to resolve the individual contributing processes locally over the entire 20$^{th}$ and early 21$^{st}$ century.

The updated GMSL extends our earlier estimate based on slightly different techniques (Dangendorf et al., 2019; the RSOI approach was used as a post-processing step at timescales smaller than 30 years, while here the RSOI is integrated directly
into the Kalman Smoother framework for the reconstruction of sterodynamic sea level) by six more years. The reconstruction shows excellent agreement to satellite altimetry in terms of variability (median r ~ 0.86 over the entire oceans), trends (median differences of 0.02±0.42 mmyr$^{-1}$), and accelerations (median differences of -0.01±0.13 mmyr$^{-2}$) over their overlapping period from 1993 to 2021. We also find moderate to good agreement between reconstructed sterodynamic sea level and independent steric height estimates from temperature and salinity fields in terms of correlation but note that three out six ocean basins show
trends that are larger than in steric height fields. Compared to tide gauge records extending back into the early 20$^{th}$ century, the reconstruction at a nearby site, as a sum of barystatic GRD, sterodynamic sea level, IBE, and GIA, often underestimates the absolute trends. However, we demonstrate that for 90 sites that are equipped with nearby GNSS the resulting trend error can be reduced by ~40% when VLM from GNSS is considered in the comparison. This indicates that unaccounted residual VLM, for instance, induced by fluid withdrawals or tectonics (Shirzaei et al., 2022), which has not been explicitly included as
a prior, is a significant driver of these differences. It also indicates that the Kalman Smoother framework is robust against such local processes.

Over 1900 to 2021 our GMSL record shows a long-term trend of about 1.5 mmyr$^{-1}$, a value that is consistent with the central estimate provided by the Intergovernmental Panel on Climate Change 6$^{th}$ Assessment Report (Oppenheimer et al., 2019; Fox-Kemper et al., 2021). Barystatic sea level and steric expansion have contributed approximately 2/3 and 1/3 to this increase,
respectively. Multidecadal nonlinear trends of GMSL rise indicate four phases of varying rates: moderate rates in the early 20$^{th}$ century (~1 mmyr$^{-1}$), enhanced rates at the end of the 1930s and the beginning of the 1940s (~2 mmyr$^{-1}$), reduced rates in the 1960s (~0.5 mmyr$^{-1}$), and a persistent acceleration thereafter. While this pattern generally agrees with previous estimates of GMSL change (Church and White, 2011; Hay et al., 2015; Dangendorf et al., 2017; Dangendorf et al., 2019; Frederikse et al., 2020; Palmer et al., 2021), the peak in the late 1930s occurs almost a decade earlier than in other reconstructions. This
earlier occurrence is more consistent with the sum of individual components from Frederikse et al. (2020) and represents primarily the result of the consideration of pre-described barystatic GRD estimates in the Kalman Smoother. More research is needed to clarify by how far uncertainties in the barystatic GRD estimates (Malles & Marzeion, 2021) may affect the budget and the Kalman Smoother estimates of total GMSL change. The persistent acceleration after the low rates in the 1960s has led



to central estimates of nonlinear rates, for the first time, exceeding 4 mmyr$^{-1}$ since 2019 at a multidecadal timescale (**Figure 10**). The exceedance of this 4 mmyr$^{-1}$ level may mark an important threshold change. For instance, it has recently been reported that widespread retreat of coastal habitats such as tidal marshes and mangroves is likely under sustained sea-level rise rates of 4 mmyr$^{-1}$ (Saintilan et al., 2023). The fact that GMSL has now exceeded that threshold means that a large portion of the ocean is already being aimpacted by those rates. While the acceleration was initiated by steric expansion in the 1960s and early 1970s, it has been the mass loss from glaciers and ice sheets that has dominated the acceleration over the last few decades. Spatially, the highest coastal trends since 1900 have occurred in the vicinity of western boundary currents and tropical regions. The acceleration since 1970 was highest in the Kuroshio region, the Subtropical North Atlantic (Dangendorf et al., 2023; Yin et al., 2023), and (sub-)tropical regions of the Indo-Pacific regions consistent with earlier studies (Merrifield et al., 2006; Dangendorf et al., 2019). In contrast to GMSL, the varying rates of sea-level rise in those regions have been dominated by sterodynamic processes at timescales of ~30 years. This reinforces our conclusion that a better understanding and prediction of local sea level at decadal timescales requires a fundamentally deeper understanding of sterodynamic variability (Dangendorf et al., 2021).

Our new reconstruction may be relevant for several applications. First, it extends the satellite record back to 1900, adding more than four times the original data. Thus, the reconstruction can be used to better characterize natural variability in local sea level and provide a more robust detection of emerging trends and accelerations from the satellite record. Trend and acceleration detection techniques have recently been used for decadal predictions/trajectories (Hamlington et al., 2022; Sweet et al., 2022) as a decision-making support tool in the realm of dynamic adaptation pathways (Haasnoot et al., 2013). The new reconstruction also provides important information on sea level and its contributing factors in otherwise data-sparse regions such as South Asia and Africa (Woodworth et al., 2010), complementing the sparse tide gauge records that exist and serving as a benchmark for datum homogenization procedures of uncertain observations (Hogarth et al., 2020). The reconstruction can also be used as a boundary condition for global (Muis et al., 2020) and regional storm tide modelling (e.g., Arns et al., 2015) to incorporate possible dependencies and changes through time or serve as a basis for high resolution sea-level reconstructions for impact assessments (Treu et al., 2023). Last, we note that an earlier version of our reconstruction (Dangendorf et al., 2019) was partially based on historical climate model outputs as priors. This prevented the IPCC from more detailed comparisons with climate model outputs over the historical period in addition to tide gauges (Oppenheimer et al., 2019; Fox-Kemper et al., 2022). The new reconstruction is entirely based on observational products and thus independent of climate model simulations. Thus, more detailed comparisons as well as detection and attribution assessments are now possible.

**Author contributions**

S.D. designed and performed the research and wrote the first draft of the paper. All authors shared ideas and contributed to the writing of the manuscript.





**Competing interests**

The authors declare that they have no conflict of interest.

**Acknowledgements**

We acknowledge Carling Hay for sharing and explaining her code, and for providing comments and suggestions on an early draft. We thank Lambert Caron and Thomas Frederikse for sharing their GIA and GRD datasets, respectively. Chris Piecuch

is acknowledged for his thorough review of an earlier draft of the manuscript. S.D., Q.S., T.W., and P.T. acknowledge the NASA grant 80NSSC20K1241. S.D. also acknowledges David and Jane Flowerree for their endowment funds.

**Financial support**

This research has been supported by the National Aeronautics and Space Administration (grant number: 80NSSC20K1241).

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
