# Peer review of "Probabilistic reconstruction of sea-level changes and their causes since 1900"

_Earth System Science Data, 2024_

## Author Comment (AC1)

Dear Editor and Reviewers,

We thank you for your positive feedback and the critical evaluation of our manuscript. We have taken each comment and suggestion into account. Below we provide our responses (in blue) to each individual comment/suggestion and describe how they were incorporated. We also note that we added a new figure 1 showing a schematic of the Kalman Smoother reconstruction process. We are confident that the revised version is significantly improved and look forward to your feedback.

Sincerely,

Sönke Dangendorf

(on behalf of all co-authors)

**Reviewer 1:**

In this paper, Dangendorf and colleagues present a reconstruction of regional and global mean sea-level change and its different drivers since 1900, based only on observations and statistical methods (to reconstruct sea level back in time). The dataset presented in this manuscript is a valuable addition to the field, and can be used in a range of scientific applications. The manuscript explains well the method and discusses some of the regional patterns of the reconstruction, displaying the multitude of processes that can be investigated with this dataset. My only concern here is that it should be made clearer to the reader that this is an update of a previous reconstruction, with some, albeit very important, alterations, so that the novelty of the dataset is not overstated. With this made it clear, and some small alterations, the manuscript deserves to be published to the scientific community.

We thank the reviewer for their positive evaluation and critical review. In the revised manuscript we have taken into account each comment. We are confident that the revised version of the manuscript is clearer in particular with respect to how this new reconstruction differs from former works (in particular Hay et al. (2015, 2017) and Dangendorf et al. (2019)).

Main comments:

- Novelty in relation to Dangendorf et al (2019).: The need to update the reconstructions available is well delineated in the introduction, which makes the issue and novelty of the study very clear. But I think it should be more explicit the differences and updates regarding the previous reconstruction from the lead author (Dangendorf et al., 2019). This is study is very briefly mentioned in the introduction, but it deserved more attention at the beginning to make it clear that this is an update/improvement of the previous reconstruction. From the introduction, that is not clear, and the reader is led to think that this is an update of Hay et al (2013; 2015). In the discussion, however, it becomes slightly more clear that this is an update of Dangendorf et al (2019) [Lines 608-610]. Only in the discussion [Lines 608-610] the authors say explicitly that this is an update of Dangendorf et al. (2019), extending the previous reconstruction by 6 years.

Thank you for pointing this out. We believe that this is a misunderstanding, resulting from vague language in the discussion. It is important to note that Dangendorf et al. (2019) combined the reconstruction techniques of Hay et al. (2013, 2015; Kalman Smoother) and Calafat et al. (2014, EOF-based) purely based on post-processing the results from Hay et al. (2015). The Kalman Smoother results were low-pass filtered and the residual signal was then used to reconstruct high-frequency variability of monthly MSL. Finally, both were added to derive an updated reconstruction. The new reconstruction presented here, however, adjusts the prior information on individual processes that is taken into account within the Kalman Smoother, but the framework itself follows that of Hay et al. (2015). Thus, the sentence in the discussion is misleading, as it is not really an extension; please note that the difference in the techniques as outlined above was already emphasized in the original manuscript. We have therefore adjusted the sentence in the discussion as follows:

"*The new reconstruction provides six more years of data than our earlier works using different techniques (Dangendorf et al., 2019; the RSOI approach was used in that study as a post-processing step at timescales smaller than 30 years, while here the RSOI is integrated directly into the Kalman Smoother framework for the reconstruction of sterodynamic sea level)*".

Adjusting the introduction to include additional discussion of Dangendorf et al. (2019) is, in this sense, unnecessary, and in our view, it would also detract from the flow of the material.

- Colormap of figures: In all sea-level literature, warm colors are related to positive trends, and cold colors to negative. The colormap used by all the authors is the opposite, which is very confusing. I really urge the authors to flip the colormap, so that blue will be negative and red positive.

  We understand the reviewers' concern but feel that this is a matter of style. Although most people in the sea level community use an opposite palette, that choice – in which blue signifies less water - is counterintuitive to non-experts (we have received this feedback in earlier published work). At the end, color bars are provided for each figure making it clear how to interpret each plot. We therefore decided to keep the current color scheme.

Minor comments:
- Not sure if there is space in the abstract, but just by reading it, it was unclear what was the 'novelty' regarding the previous reconstructions.

  We have adjusted the abstract to highlight the novelty of the study. Line 16-18 now reads as follows:

  "*The reconstruction is based on tide gauge records and incorporates prior knowledge about physical processes from ancillary observations and geophysical model outputs allowing us, for the first time, to resolve individual processes and their uncertainties.*"

- L21: make it clear that the reconstructions were larger (though with larger reconstructed trends in three out of six regions).

We adjusted the sentence as follows:

*"Validation against steric height estimates based on independent temperature and salinity observations over their overlapping periods shows moderate to good agreement in terms of variability, though with larger reconstructed trends in three out of six regions."*

- L24: "most recent rates"… unclear to what period this refers. Please give a year.

  Thank you, done

- L49: Is there a more recent GMSL acceleration that could be mentioned (instead of a 2018 publication)?

  We do not know any peer-reviewed publication providing an update, but we also feel that Nerem et al. (2018) is the original work, and no further acceleration has been noted since.

- L127: I'm not so familiar with these methods, but how are the tide-gauge observations, which is in RSL, and satellite altimetry, which is ASL, combined in the state vector?

  They are not. The state vector contains sea level **estimated** at those locations not the actual observations.

- L154: does this mean than that everything before 1955 is only based on statistical modelling?

  It means that we assume that the covariance matrix is properly represented by data after 1955 (i.e., the information on how individual tide gauges covary). This is a necessary tradeoff for calculating covariances of gappy data. However, it does not mean that everything before 1955 is only based on statistical modelling, as some tide gauges extend further back in time.

- L161 (and L203): Why only one IBE estimate? Were there no other reanalysis available, to have a larger number, like for GRD and GIA…? And this IBE estimate is the combination of two reanalysis (one 1900-2015 and one 2016-2021), were there no single one that covered the entire study period? Do these two dataset have any significant differences (apart from period)?

  The reason is two-fold. First, the 20th century reanalysis is the only remaining long-term reanalysis, while ERA20C is no longer available. Second, any additional dataset would double the ensemble size and make the computation very inefficient. Given the fact that the IBE is only a minor component within the analysis, we have included only one member. We also note that the IBE was often ignored in other reconstructions, so our methodology is still an advancement compared to other reconstructions. We have added this reasoning to the data section, which now reads as follows:

  *"IBE variations are derived from sea-level pressure data from atmospheric reanalysis models, specifically the 20th century reanalysis v3 covering the period 1900 to 2015 (Slivinski et al., 2019) and the NCEP-NCAR reanalysis 1 (Kalnay et al., 2018) from 2016 to 2021. Both reanalyses were interpolated from different global grids onto the 74,742 grid points for the regional sea-level fields considered in the Kalman Smoother. They are combined after adjusting them to the same mean at each grid point over the five overlapping years between 2011 and 2015. The choice of using the 20th century reanalysis v3 is based on its availability over long periods. Other datasets (e.g., HADSLP) are not considered, as the IBE has only a very minor contribution compared to*

*other processes and every additional dataset would double the total ensemble size considered here.*"

Lastly, we note that differences between the different data sources may occur in data-sparse regions (usually at highest latitudes away from tide gauges). That's why we chose a short overlapping period for their vertical adjustment to avoid any step changes in the combined dataset. Again, given the minor importance of the IBE, we feel that any more detail is beyond the scope of this manuscript.

- L191: Why only 5 years were considered to extrapolate the last year of GRD?

  We used only 5 years as the ice-contributions have accelerated and are generally very nonlinear. Using a longer period would not account for that. However, we note that we extend by only one value and, as already noted in the original manuscript, using a quadratic fit instead does not yield different results.

- L205: Use the grid resolution instead of number of grid cells (same for other occasions when the number of grids is mentioned (e.g., Section 4)). For example, L222: Does this mean that you at the end have a 4-degree resolution dataset?

  It is not a fully gridded product, as we deleted land areas (NaNs) and selected every fourth grid point from satellite altimetry. As satellite altimetry has a resolution of ¼ degree, we end up, on average, with a 1-degree resolution. However, we did not interpolate it back onto an equal-spaced grid. This can be done by any user depending on their need.

- L213: Satellite altimetry was also used as input into the state matrix of the Kalman filter, no?

  No, it was only used as input for the RSOI reconstruction of sterodynamic residuals. It determined the number of grid points on which the whole reconstruction was performed though.

- Validation with satellite altimetry: If altimetry was used in the Kalman filter, is this a fair validation?

  Satellite altimetry covariance information was used to determine the sterodynamic component, so it is not fully independent. This is why we perform the comparison to independent steric products below. Unfortunately, there is no other way of validating those products to observations. However, we emphasize that the skill in reproducing satellite altimetry was significantly improved compared to earlier attempts summarized in Carson et al. (2017), which illustrates that the good agreement is not assured with the use of covariance information from satellite altimetry.
  We have added a note that makes it clearer that they are not fully independent:

  "*The GMSL and basin-scale averages (based on basin definitions following Thompson & Merrifield, 2014) of the new reconstruction of relative sea level and its individual contributors are presented in **Figure 3**. Overlaid are the global and regional averages from satellite altimetry. We note that covariance information from satellite altimetry was used to determine the sterodynamic component in the reconstruction, and as such the validation here is not fully independent.*"

- L231: A suggestion, but you could use this value (r=0.89), or another as a reference contour line in the global map (Figure 2a).

  We tested this, but it becomes very messy, as the correlation decreases primarily in eddy-rich areas, which are then overlaid by black lines. We therefore decided to retain the version that we had in the original manuscript.

- L241: Does this mean that there is a resolution limit to the reconstruction?

  Yes, this is what was incorporated in the prior sentence ("*The reconstruction is also meant to capture primarily large-scale patterns by reducing it to the 10 major modes*"), although we cannot specify the actual limit here. This needs to be explored by users based on their application.

- L255: Is there a way of indicating which regions that have a significant residual? If hatching all the areas that are insignificant would make the figure "too dirty", maybe you could mention in the text, since its such a small percentage, where the residuals are significant?

  We tried highlight such regions, but they are invisible because they are so few. They are also not clustered in one specific region. We have added a note:

  "*The majority of trends (98.9% of the entire ocean area) and accelerations (96.5% of the entire ocean area) in the residuals (satellite observations minus reconstruction) are also not statistically significant, and the few that are statistically significant do not cluster in a specific region.*"

- Figure 3: The differences (last row) in the accelerations seems much larger, but this is probably an artefact of the color range. Maybe for these panels, both the trend and acceleration should be on similar color range, or the differences should be normalized.

  The acceleration coefficients are more sensitive to interannual variability, and this is why they show (relatively) larger magnitudes. Acceleration coefficients should be treated with far more care than linear trends (see Haigh et al., 2014 for a discussion: https://www.nature.com/articles/ncomms4635). That's why we feel that the presentation is OK as is.

- Sterodynamic estimates: There are some regional budgets, such as Wang et al (2021), Royston et al (2020) and Camargo et al (2023), that do have sterodynamic estimates. So these could be used to validate the sterodynamic of the reconstructions, instead of only independent steric estimates.

  The budgets from Royston only cover the period from 2005 to 2015. Camargo et al. (2023) uses different basin definitions than we do (that cannot be easily merged). Also, the data from both publications overlap with that represented by Frederikse et al. (2020). Wang et al. (2021) combine a global estimate that was included in Frederikse et al. (2020) and couple it with an ocean reanalysis, so it is not purely observational. Ocean reanalyses come with their own drifts and biases (e.g., Dangendorf et al., 2021), which was also one of the reasons why they were not considered as an appropriate prior for the Kalman Smoother. We also note that the authors never made that field available (only trends over the period they assessed), so it cannot be added to this figure. Lastly, we note that Frederikse et al. (2020) incorporated multiple observational products, which motivates our choice to focus on that product.

- L283: Here and also L287 you say 'Northwest Pacific', but on the table 1 and 2 is only 'West Pacific'.

  Thank you for spotting this, we adjusted the tables accordingly.

- L238: The agreement of the reconstruction is better with satellite altimetry than with tide gauge records, reflecting the dependence of the reconstruction on the quality of the altimetry in the coastal zone. In addition to the processes you mention here that altimetry measurements might misrepresent in the coastal zone, there are also a large number of correction issues in the coastal zone (see Vignudelli et al, 2019, for example). If the reconstruction is made with 'standard' altimetry product, and not dedicated coastal products (e.g., X-TRACT (Birol et al., 2017), ALES+ (Passaro, 2018)), it will inherit these issues. So, does this mean that the user should take care when using this product in the coastal zone (and say, not consider the most inshore grids?) In this case, a sentence saying this should be added. On the same note, I was wondering: (1) could the reconstruction be improved by including coastal altimetry products? (2) would the validation with the 'virtual tide gauges' (Cazenave et al., 2022) give better results than the validation with the standard tide gauges?

- We added a note on the degradation in quality due to correction uncertainties in the coastal zone:

  "*The latter may miss some fundamental processes that take place close to the shore (e.g., Cazenave et al., 2022) such as coastally trapped signals from wind (Calafat and Chambers., 2013) and river discharge (Piecuch et al., 2018) or nonlinear VLM rates (Oelsmann et al., 2023), and is also known to have degraded quality due to correction issues in the coastal zone (e.g., Passaro et al., 2018; Vignudelli et al., 2019).*"

  Regarding point 2: Yes, coastal altimetry could, in principle, be included, though that would decrease the period over which we perform the EOF analysis as those coastal products are only available since 2002. The implementation will be part of a project that was recently funded, so it will be incorporated in future releases.

- L353: here you mention '64%' and '33%' of the tide gauges…what about the other 3%?

  This is based on the 1mm/yr and 0.5mm/yr thresholds in that same sentence, so there is no other 3%. It simply tells that 64%/33% of all sites show trend differences below these values. The numbers should not be added up.

- L355-358: these are also locations with large non-linear (and non-GIA) VLM, such as earthquakes hot spots (U.S. west coast) and land subsidence (Gulf of Mexico). Could this also add to the large differences?

  This is exactly what we meant. We added a note to make it clearer:

  "*Second, there may be processes that affect tide gauges at the coast (such as nonlinear VLM) but that are not explicitly included as priors in the Kalman Smoother fields*"

- L364: Could add a comment about why the differences with the tide gauges have so much larger uncertainties than the differences with satellite altimetry?

  Not sure what is meant here? The remainder of the paragraph deals with precisely this point. We hypothesize that there are additional local processes, such as VLM, that lead to larger differences between the reconstruction and tide gauges.

- L377-384: This part was very confusing for me, and required me to read several times to understand the different what trend values and number of tide gauges (90, 516, 119) were referring to. For example, the 0.27±1.67 and 0.1±1.8 mm/yr trends, are these average of the 90 and 516 locations? And the difference refers to KS reconstructed with GIA – tide gauges?

  We are sorry for any confusion. Yes, the numbers refer to the median ± standard deviation over all sites for the trend differences. We added that information at the beginning of the paragraph. We also added the number of sites to one sentence, which now reads as follows:

  "*However, replacing the GIA crustal rates with the GNSS-based VLM at the 90 sites significantly reduces the trend differences to -0.14±1.00 mmyr$^{-1}$ (i.e., a reduction by 40% in the standard error).*"

  The numbers mentioned by the reviewer were already linked to the number of sites, so we do not know what to change there (*The trend differences for the **90** selected sites, **0.27±1.67 mmyr$^{-1}$**, are not significantly different to those obtained at all **516** sites (**0.1±1.8 mmyr$^{-1}$**)*).

  The idea here is that unaccounted residual VLM (i.e., non-GIA) explains a part of the large differences. Thus, we replace GIA by GNSS estimates before the comparison.

- L400-402: Please add the periods of Church & White (2011) and jevrejeva (2014), instead of just say 'for shorter periods'. And add uncertainties to these trends, if provided.

  Done, thank you.

- Table 3: You could also add the acceleration rates to the table (so similar to Table 1)

  Thank you, but we decided against it as we only discuss the acceleration coefficients for the period after 1970, as outlined in the manuscript. Adding them to the table would require adding an additional text paragraph. Furthermore, the regional accelerations are already discussed in terms of the high-resolution maps.

- L424 (and Fig 8): Why is the comparison made with D19 only from 1930-1939? And in the discussion (L612-614) the results are compared for 1993-2021?

  This results from the discussion in the paragraph. The reconstructions disagree with the sum of individual components during this period. That's why we mapped the differences between Dangendorf et al., (this study) and Dangendorf et al. (2019, which is in line with the other reconstructions) over this particular period. Lines 612-614 in the discussion section refer to the comparison of the reconstruction with satellite altimetry, so we are confused how the reviewer connected the two!?

- L428-429: This information should be given before, either in the introduction or methods.

  As noted in our response to the main comment, this is not an update of Dangendorf et al. (2019), so methodological details on this study are not relevant to any other section. It was highlighted here as it shows similar global behavior as other reconstructions and as it is the only reconstruction for which regional fields are available.

- In Figure 10 you compare the accelerations with Palmer et al (2022), but this comparison never comes up in the text.

*Thank you, we added a note. Text now reads as follows:*

*"The increase in the rates since the early 1970s (which is consistent with former reconstructions summarized in Palmer et al. (2021)) was initially driven by the steric contribution and later by the barystatic component of sea-level rise."*

- L497: And to a smaller extent also due to GRD contribution

*Thank you, we added that information, which now reads as follows:*

*"Negative trends are primarily located in centers of postglacial uplift where former ice sheets were located (Scandinavia and North America) and are therefore caused by the GIA signal and to lesser extent due to GRD in response to modern melting (**Figure 12d**)"*

- L515: It seems like there is something missing in this sentence "The local maxima result from additional around…"

*Thank you, we corrected the sentence as follows:*

*"This pattern is largely dominated by the barystatic GRD terms and results from the fact that most major melt-sources in the 20$^{th}$ century were in high (northern) latitudes. The local maxima around ~40 degrees North and South are due to the sterodynamic contribution and reflect gyre dynamics."*

- L590: Are the codes ready for publication already? Or does this mean that there will be another publication for the code? It was unclear.

It will only be published with a future release and publication. The publication of the codes and all prior data requires considerable data management resources that were not provided with the funding for this project. However, in the follow up project that was just approved those efforts will be funded, and accordingly we will prepare the codes in a way that they are publicly sharable. We added a note that makes it hopefully clearer:

*"The code for the Kalman Smoother reconstruction is currently still being prepared for publication but available from the main author upon request. We intend full publication of the Kalman Smoother approach with the next version of the reconstruction in a future release with cleaned-up and annotated codes."*

- L630: It's a bit circular that the peak matches better with the ones from Frederikse et al (2020), since the barystatic GRD used in the reconstruction is from Frederikse et al (2020).

Not necessarily. If it was circular, they should show up in the Frederikse et al. (2020) tide gauge reconstruction as well (tide gauges were corrected for the same barystatic priors as used here), but they do not.

Editorial comments:
- L33: add comma after warming climate.

Thank you, done

- L36: add comma after parenthesis.

Thank you, done

- L50: add comma after parenthesis.

Thank you, done

- L78: "unrelated through GIA": should this 'through' be a 'to'?

  Thank you, done

- L92: Add comma after observations.

  Thank you, done

- L102: I always see only 'IB' instead of 'IBE'…

  We have seen it both ways, so we keep it as is.

- L107: change 'or' to 'and'

  Thank you, done

- L108: 'MSL' is introduced here, but not used again afterwards…

  removed

- Equation 2: first symbol on the right-hand side is not introduced/explained in the text.

  Thank you, for spotting this. We adjusted the equation

- L128: modification instead of adjustment? (adjustment has a more negative connotation, seems like the previous work was wrong). And same for L129.

  Thank you, done

- Equation 3: Should the 'and' be in a new line?

  Thank you, done

- L166: use VLM, which was already introduced.

  Thank you, done

- L171: should it be 4000 (L162)?

  Thank you, done

- Figure 1a: Some of the symbols on the dark red and blue are a bit hard to see. Maybe using lighter shades for the background colors might improve it.

  Thank you, done

- L275: Reference should be Camargo (2020).

  done

- L349: Change 'Fig.' to Figure (as it has been in all other instances)

  Thank you, done

- L381: Here you have '-0.14±1.00 mm/yr', but Fig 6c says -0.13.

  adjusted

- L400: Here you have 1.5±0.19, but Table 3 is 1.5±0.20.

  adjusted

- L501: I would suggest using 'scarcity' instead of 'paucity', but that's because I had to look up this word.

Was suggested by a native speaker, so we'll keep it as is.

References

- Royston, S., Vishwakarma, B. D., Westaway, R., Rougier, J., Sha, Z., and Bamber, J.: Can We Resolve the Basin-Scale Sea Level Trend Budget From GRACE Ocean Mass?, J. Geophys. Res.-Oceans, 125, 1–16, https://doi.org/10.1029/2019JC015535, 2020.

- Wang, J., Church, J. A., Zhang, X., Gregory, J. M., Zanna, L., and Chen, X.: Evaluation of the Local Sea-Level Budget at Tide Gauges Since 1958, Geophys. Res. Lett., 48, 1–12, https://doi.org/10.1029/2021GL094502, 2021.

- Camargo, C. M. L., Riva, R. E. M., Hermans, T. H. J., Schütt, E. M., Marcos, M., Hernandez-Carrasco, I., and Slangen, A. B. A.: Regionalizing the sea-level budget with machine learning techniques, Ocean Sci., 19, 17–41, https://doi.org/10.5194/os-19-17-2023, 2023.

- Vignudelli, S., Birol, F., Benveniste, J. *et al.*Satellite Altimetry Measurements of Sea Level in the Coastal Zone. *Surv Geophys* **40**, 1319–1349 (2019). https://doi.org/10.1007/s10712-019-09569-1

- Birol, F., N. Fuller, F. Lyard, M. Cancet, F. Niño, C. Delebecque, S. Fleury, F. Toublanc, A. Melet, M. Saraceno, F. Léger, 2017. "Coastal Applications from Nadir Altimetry: Example of the X-TRACK Regional Products." Advances in Space Research, 2017, 59 (4), p.936-953. doi:10.1016/j.asr.2016.11.005

- Passaro, Marcello, Stine Kildegaard Rose, Ole B. Andersen, Eva Boergens, Francisco M. Calafat, Denise Dettmering, Jérôme Benveniste, ALES+: Adapting a homogenous ocean retracker for satellite altimetry to sea ice leads, coastal and inland waters, Remote Sensing of Environment, Volume 211, 2018, Pages 456-471, ISSN 0034-4257, https://doi.org/10.1016/j.rse.2018.02.074.

- Passaro, M. et al. ALES: A multi-mission adaptive subwaveform retracker for coastal and open ocean altimetry. *Remote Sensi. Environ.***145**, 173–189 (2014).

- Cazenave, A., Gouzenes, Y., Birol, F. *et al.*Sea level along the world's coastlines can be measured by a network of virtual altimetry stations. *Commun Earth Environ* **3**, 117 (2022). https://doi.org/10.1038/s43247-022-00448-z

---

## Author Comment (AC2)

Dear Editor and Reviewers,

We thank you for your positive feedback and the critical evaluation of our manuscript. We have taken each comment and suggestion into account. Below we provide our responses (in blue) to each individual comment/suggestion and describe how they were incorporated. We also note that we added a new figure 1 showing a schematic of the Kalman Smoother reconstruction process. We are confident that the revised version is significantly improved and look forward to your feedback.

Sincerely,

Sönke Dangendorf

(on behalf of all co-authors)

**Reviewer 2:**

This paper provides a new probabilistic reconstruction of historical global and regional sea-level changes by combining two methods (RSOI and Kalman smoothing) and using recently published results to constrain the reconstruction. Overall, the paper is clearly written, the methods used are well explained (although some improvements could be made I think - see below) and the results bring new information on historical sea level, which makes this paper and the related dataset definitely of interest for future research and operational studies. I only have a list of minor comments, suggested corrections and questions that I have listed below.

We thank the reviewer for their positive and critical evaluation of the manuscript.

- Please include more carriage return in the introduction to make it easier to read.

  The template format might have given the impression that parts of the manuscript are one long paragraph, while in reality they are not. In the final print lines 47-97 should appear as 3-4 paragraphs

- P2l48: 3.4 mm/yr over which period ? please clarify.

  Added (since 1993)

- P4l122: "control input parameter" or "control input parameters" ?

  Thank you, corrected

- P4l123: please revise the equations (1) and (2). The "k" subscript on the right hand side should be replaced by "k-1" in (1) and the "O with stroke" in (2) should be replaced by phi (state transition matrix) I presume.

  Thank you for spotting this, corrected!

- P4l128: "the latter […]" please reformulate. I don't really understand what is the first adjustement compared to Hay et al.

  We adjusted the sentence, which now reads as follows:

*"The reduction of the vector to only one globally uniform GMSL estimate is the first modification to Hay et al. (2015, 2017), where the vector also contained estimates of 21 melt rates from mountain glaciers and ice sheets."*

- P4l130-132: were all individual GRD estimates used from Frederikse et al. (2020) or the total contribution (i.e. 1 GRD estimate for all components)? Please clarify.

We added "*total*", but we also note that the selection of the prior estimates is discussed in the data section. There we also added a sentence "*Here, the sum of the individual contributions is considered.*"

- P6l171: replace 4,0000 by 4,000

Thank you, done!

- Although this not done either in Hay et al. (2017), I'd suggest to add a flowchart Figure that illustrate the various steps of section 2.1 to guide the reader through the methodology and also underline in this flowchart, where the present work differs from Hay et al. (2017) (and Dangendorf et al., 2019) to better capture the novelty.

Thank you, we added a schematic as Figure 1 that illustrates the main equation in combination with the data that goes into them. We do not see how differences to Hay et al. (2015) and Dangendorf et al. (2019) can be easily incorporated. Those, however, have been clearly outlined in the text.

- What is the time step of the procedure? annual? Please clarify.

This information is provided in the introduction as well as the next sentence (first sentence of data section), so we decided to keep this as is.

- P7l213: " […], which we interpret as sterodynamic" (?)

Not sure what the issue here was but we adjusted the sentence as follows:

*"The resulting sterodynamic sea-level fields are then used as input parameters for the Kalman Smoother."*

- Figure 2: please recall in the caption the time period over which correlation is calculated for clarity.

Thank you, done

- Figure 4: I've the impression that the mismatch between the present reconstruction and Frederikse et al. (2020) is well pronounced before ~1990, while from 1990-onwards, the comparison seems good. Could that be the effect that the RSOI-based reconstructed sterodynamic component is strongly constrained by the availability of altimetry data? Please comment.

The two curves match very well over most of the record with the exception of the behavior in the 1930s and 1940s (the rest is largely spurious local variability from the reconstruction process which is larger in Frederikse et al. (2020) due to the averaging technique that they used). Both techniques are not directly comparable. Frederikse et al (2020) is based on a virtual station technique (i.e., averaging) and does not use any satellite altimetry, so we cannot see how the use of satellite altimetry should influence their (dis-)similarity. We also note that it is only covariance information that comes from

satellite altimetry, while tide gauges are still the main source of information in the reconstruction.

We had a global sea level reconstruction intercomparison project a few years ago (https://www.issibern.ch/teams/unifysealevel/) in which we tested the different reconstruction techniques in terms of their performance in climate model fields in which the model truth is a priori known. The conclusions from that intercomparison (not yet published) were that the handling of VLM information was the main source of uncertainty between different techniques (which is in line with results presented in Dangendorf et al., 2017) with advantages towards field reconstructions (virtual station techniques can be strongly biased by individual gauges with datum shifts or other local processes, that would be counted as outliers in the field reconstructions).

- P14l340-342:"This limitation is evident […] smaller amplitude signal in the tide gauge record of San Fransisco". This statement is unclear to me. I rather see a larger amplitude signal in the TG record (?). Also, figuring where the 1997/1998 El Nino signals pops-up is not straightforward.

We adjusted the text as follows:

"*This limitation is evident by comparing satellite altimetry observations and the reconstruction during individual peaks such as the 1997/1998 El Nino where the reconstruction underestimates the amplitude compared to the tide gauge record at San Francisco (Figure 6a)*"

- P15l367-369: GNSS datasets used should be described in section 2.2 (Data). I supposed the ULR7A solution from Gravelle et al. (2023) was used? If yes, please clarify this too (as four solutions are available on SONEL actually).

Yes, we used URL7A and added that information. We only included data for the reconstruction itself in the data section and introduced validation data (as GNSS and the other steric products) in the main text. This ensures that the text remains brief and avoids any unnecessary double mentioning.

- P15l381: -0.14 or -0.13 ? Please make the main text and Figure 6c consistent.

Done, thank you

- P17l415: Maybe it would be relevant to cite the corresponding Figure of Frederikse et al. (2020). Fig1c right ?

The record is shown in Figure 7b (now Figure 8), so it is unclear why the Figure in Frederikse et al. (2020) should be cited.

- Figure 7: colors in the plot and the legend for Frederikse (2020) budget don't match. Please correct this.

Corrected, thank you!

- P18l445: please refers to Figure 9 by quoting Figure 9 already there.

Done, thank you!

- Figure 9: the Sterodynamic component rates suggests ~30yrs oscillation. Is that possibly related to climate variability or is it related to the reconstruction processing? Such an oscillation is not obvious in e.g. Frederikse et al (2020) (see their Figure 1c)

This is hard to tell given the huge differences between individual observational products and the large uncertainties in that component within our reconstruction. For the moment we prefer not to emphasize any of that given the reconstruction uncertainties.

- P21l473: "2.2 +/- 0.11" while Table 3 gives 2.2 +/- 0.12. Please make the main text and Table information fully consistent.

Done, thank you

- P22l477-479: here also main text and Table 3 rates estimates are not always consistent. Please adjust

Thank you, we forgot to update the numbers in the text but have adjusted them now.

- P23l495-496: Figure 11a colorscale does not allow appraising such larges ranges as it ranges from -3 to 3 mm/yr. Maybe modify the caption or adjust the main text to clarify this.

Thank you. The difficulty is to emphasize the main features in the different contributions. As a solution, we added information to the colorbars in Figure 11 and 12 (now 12 and 13) that the values can exceed or fall below the color range.

- P23l497-498: "are therefore by the GIA signal". A word is probably missing (?).

Thank you, we adjusted the sentence as follows:

"*Negative trends are primarily located in centers of postglacial uplift where former ice sheets were located (Scandinavia and North America) and are therefore caused by the GIA signal and to lesser extent due to GRD...*"

- P23l515-516: "The local maxima […] reflect gyre dynamics". unclear. Reformulate ?

We adjusted the sentence as follows:

"*The local maxima around ~40 degrees North and South are due to the sterodynamic contribution and reflect gyre dynamics.*"

- P24l519 : Would it be worth and reasonably feasible to test a different SLP dataset ? E.g. HadSLP? or in light of what is done for GIA, using various ensemble members of the 20CR reanalysis dataset?

The choice of using only one SLP dataset was motivated by the fact that an additional dataset would double the ensemble size, which makes the computation too heavy. We also note that the overall contribution by inverse barometer effects is very small compared to the other components. Therefore, we decided to stick to one dataset in this version but aim at testing multiple datasets in future releases. Please also note the response to a very similar comment made by the other reviewer.

We have adjusted the methods section, which now reads as follows:

"*IBE variations are derived from sea-level pressure data from atmospheric reanalysis models, specifically the 20$^{th}$ century reanalysis v3 covering the period 1900 to 2015 (Slivinski et al., 2019) and the NCEP-NCAR reanalysis 1 (Kalnay et al., 2018) from 2016 to 2021. Both reanalyses were interpolated from different global grids onto the 74,742 grid points for the regional sea-level fields considered in the Kalman Smoother. They are combined after adjusting them to the same mean at each grid point over the five overlapping years between 2011 and 2015. The choice of using the 20$^{th}$ century*"

*reanalysis v3 is based on its availability over long periods. Other datasets (e.g., HADSLP) are not considered, as the IBE has only a very minor contribution compared to other processes and every additional dataset would double the total ensemble size considered here.*"

- P28l608-610: this information should be made very clear in the introductory part too I believe.

  We added that information also to the end of the introduction, which now reads as follows:

  "*Second, the approach enables the estimation of the individual contributors and sea-level change as constrained by tide gauges. As such we derive a novel global and regional mean sea level reconstruction at annual resolution covering the period from 1900 to 2021 extending the reconstructions from Hay et al. (2015) and Dangendorf et al. (2019) by 11 and 6 more years, respectively*"

- P29l638 typo 'aimpacted'

  corrected